**RESEARCH**

# Most human DNA replication initiation is dispersed throughout the genome with only a minority within previously identified initiation zones

Jamie T. Carrington[1†], Rosemary H. C. Wilson[1,4†], Eduardo de La Vega[2], Sathish Thiyagarajan[2], Tom Barker[2], Leah Catchpole[2], Alex Durrant[2], Vanda Knitlhoffer[2], Chris Watkins[2], Karim Gharbi[2] and Conrad A. Nieduszynski[2,3*]

†Jamie T. Carrington and Rosemary H. C. Wilson equal contribution.

*Correspondence:
conrad.nieduszynski@earlham.ac.uk

[1] University of Oxford, Oxford, UK
[2] Earlham Institute, Norwich Research Park, Norwich NR4 7UZ, UK
[3] University of East Anglia, Norwich, UK
[4] Cardiff University, Cardiff, UK

## Abstract

**Background:** The identification of sites of DNA replication initiation in mammalian cells has been challenging. Here, we present unbiased detection of replication initiation events in human cells using BrdU incorporation and single-molecule nanopore sequencing.

**Results:** Increases in BrdU incorporation allow us to measure DNA replication dynamics, including identification of replication initiation, fork direction, and termination on individual nanopore sequencing reads. Importantly, initiation and termination events are identified on single molecules with high resolution, throughout S-phase, genome-wide, and at high coverage at specific loci using targeted enrichment. We find a significant enrichment of initiation sites within the broad initiation zones identified by population-level studies. However, these focused initiation sites only account for ~20% of all identified replication initiation events. Most initiation events are dispersed throughout the genome and are missed by cell population approaches. This indicates that most initiation occurs at sites that, individually, are rarely used. These dispersed initiation sites contrast with the focused sites identified by population studies, in that they do not show a strong relationship to transcription or a particular epigenetic signature.

**Conclusions:** We show here that single-molecule sequencing enables unbiased detection and characterization of DNA replication initiation events, including the numerous dispersed initiation events that replicate most of the human genome.

**Keywords:** Replication origin, Origin mapping, DNAscent, Ultra-long, nCATS

## Background

DNA replication is a fundamental cellular process that is conserved throughout life. It is critical for genomic stability that genomes are replicated once and only once. In eukaryotes, DNA replication is initiated from multiple sites along each chromosome, for example, tens-of-thousands across the human genome. These sites have been linked to disease as sites of higher mutation rates [1, 2] and also implicated in chromosomal translocations [3–5]. In *Saccharomyces cerevisiae*, sites of DNA replication initiation are defined by a sequence motif bound by the origin recognition complex (ORC) [6], although they vary in how frequently they are used [7]. However, in metazoans, ORC has weak sequence specificity and there is conflicting evidence for sequence bias, such as G-quadruplexes, at replication initiation sites [8–10].

Numerous genomic methods have been used to examine either the dynamics of DNA replication or to directly identify the sites of replication initiation [10]. Predominantly, these methods utilize short-read DNA sequencing (or microarrays) and measure the average signal from a population of millions of cells. Cell population replication dynamics have been studied either by examining replication time (e.g., repli-seq [11, 12]; sort-seq [13, 14]) or by determining replication fork direction (e.g., Okazaki fragment sequencing—"Ok-seq" [15, 16]; Polymerase-usage sequencing—"Pu-seq" [17, 18], GLOE-seq [19]). In mammalian cells, these methods have identified broad zones of replication initiation (30–100 kb, initiation zones—IZs), separated by large regions (interquartile range 83–369 kb; median 183 kb) of implied unidirectional fork progression [16]. There is a high degree of concordance within and between these approaches, for example, in the identified IZs [10]. Alternatively, multiple approaches have been used to directly identify replication initiation sites, including the abundance of short nascent strands (SNS-seq) [20], "bubble" structures at activated origins [21], early replication (e.g., EdUseq-HU; ini-seq) [22–24] or binding of replication initiation factors (chromatin-immunoprecipitation of ORC or MCM complexes) [25–27]. However, in mammalian cells, these approaches differ in the number and locations of sites identified and show poor concordance with IZs identified from replication dynamics studies [10, 28, 29]. This discrepancy could result from each assay detecting different steps in the process of genome replication (e.g., licensing, origin activation, fork progress). However, it has also been postulated that this inconsistency could be due, in part, to the reliance on cell population-based approaches to study what may be a heterogenous process, thus resulting in low sensitivity, and a high rate of false-negatives [10].

Single-molecule and single-cell analyses of DNA replication have the potential to resolve the complexity of heterogeneity within cell population data. Genome sequencing from single cells in S phase reveals which portions of the genome have already replicated [30]. This gives a "snapshot" of the state of genome replication, but without the resolution to identify individual replication initiation sites. By comparison, single-molecule approaches, such as combing and DNA fiber, rely on labelling nascent strands with radiolabelled nucleotides or more recently, halogen- or fluorophore-labelled analogs that are then detected by radiography or microscopy [31–33]. These methods indicate that replication initiation is highly stochastic with sites ~100 kb apart [33–35]. However, they are low throughput and generally lack genomic location information. Hybridization probes can give some location information; however, this further reduces throughput

[36]. Recently, an Optical Replication Mapping (ORM) method generated genome-wide replication dynamics on single megabase-length molecules at high coverage and with genomic coordinate information [37]. However, ORM requires cell synchronization and transfection with bulky fluorophore-labelled nucleotides and is therefore only suitable for certain cell types. In addition, ORM only labels the first 2% of S-phase, depends upon activation of the intra-S-phase checkpoint, and has a resolution of ~ 15 kb.

To address the limitations of population-sequencing approaches and optical single-molecule techniques, we developed the first genome-wide DNA sequencing method using ultra-long single molecules to detect DNA replication dynamics, called DNAscent [38]. This method utilizes nanopore sequencing and bespoke base-calling models to identify, at base resolution, the sites of BrdU-incorporation [38, 39]. We validated this approach in *Saccharomyces cerevisiae* by in vivo incorporation of BrdU during a single S-phase. Resulting patterns of BrdU incorporation allowed high-throughput and high-resolution identification of replication fork direction and sites of replication initiation, termination, and fork pausing on single molecules. This demonstrated that, even in *S. cerevisiae* with its generally well-defined origin sites, 10–20% of replication initiation events are at sites not detected by population-level approaches. Variants of this nanopore sequencing-based method have allowed genome-wide measurement of mean fork velocity [40] and ensemble identification of replication fork pause sites [41].

Here we apply DNAscent to cultured human cells and, for the first time, identify DNA replication initiation events on single, sequenced molecules across the human genome. The sites we identify are enriched within the initiation zones previously identified by population-level replication dynamics studies; we term these "focused initiation sites". However, the majority of initiation events identified by DNAscent are outside previously reported initiation zones; we term these "dispersed initiation sites". Unlike focused sites, dispersed sites are not related to a particular epigenetic mark or transcription context. We propose a model that integrates the focused sites of high replication initiation efficiency and low efficiency dispersed initiation sites occurring throughout most of the genome.

## Results

### Nanopore detection of BrdU in human genomic DNA

We set out to establish suitable growth conditions for cultured human cells (HeLa-S3 and hTERT-RPE1) that would allow detection of BrdU incorporated into nascent DNA using nanopore sequencing (DNAscent). Asynchronously growing cells were treated with a range of BrdU concentrations (0.3–50 μM), either for the duration of one cell cycle (20 h HeLa-S3 or 27 h hTERT-RPE1) or for pulses of 2 or 24 h (Fig. 1; Additional file 1: Fig. S1). Then, extracted genomic DNA was subjected to PCR-free nanopore sequencing followed by detection of BrdU at single-base resolution. This produces a BrdU probability at every thymidine position on each single-molecule read (Fig. 1A). Analysis of these BrdU probabilities showed that they have concentration-dependent bimodal distributions, each with a peak close to zero (i.e. thymidine calls) and a second peak approaching a probability of one (i.e., BrdU calls) (Fig. 1B; Additional file 1: Fig. S1 A). As expected, the proportion of high BrdU probabilities depended on the BrdU concentration used in labelling. Similar results were observed for both cell lines and various

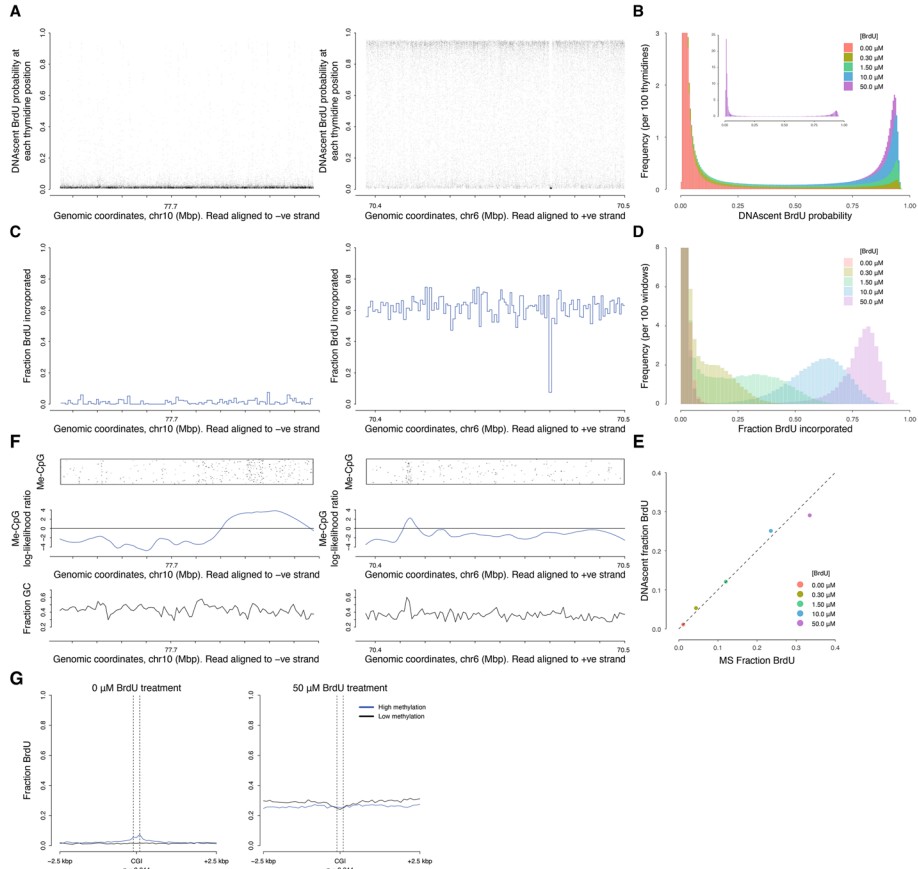

**Fig. 1** Detection of BrdU in human genomic DNA by nanopore sequencing over a wide range of BrdU concentrations. **A** Plots show two example nanopore sequencing reads from an asynchronous HeLa-S3 cell culture treated with 10 μM BrdU for 20 h. Reads are shown aligned to the human genome with DNAscent calculated BrdU probabilities for each thymidine position. Tick marks are 10 kb. **B** Frequency distribution of BrdU probabilities at every thymidine position for nanopore sequencing reads from HeLa-S3 cells grown in the indicated range of BrdU concentrations for 20 h. Insert shows the full *y*-axis for the 50 μM sample. **C** Two plots of the fraction BrdU incorporated ($p \geq 0.5$; independent windows of 290 thymidine positions) for the example nanopore reads in Fig. 1 A. Tick marks are 10 kb. **D** Frequency distribution of the fraction BrdU incorporated in each window (290 thymidine) for same reads as in Fig. 1B. **E** Comparison of fraction BrdU in DNA as determined by nanopore sequencing and mass spectrometry from the same DNA samples (as in Fig. 1B, D). Dashed line is $y = x$. **F** CpG methylation analysis for the two example reads in Fig. 1 A, C. Top row: Jitter plot of individual methylated CpGs determined by Nanopolish. Middle row: smoothed density of CpG methylation across the two reads. Bottom row: fraction GC in 1-kb windows. **G** Meta-analysis of fraction BrdU detected by DNAscent at thymidine positions relative to CGI centers for reads from HeLa-S3 cells treated with 0.0 or 50 μM BrdU (as per Fig. 1B, D). Fractions are calculated from 100 -bp windows. Dashed lines (200 bps apart) indicate the minimum bounds of CGIs. CGIs were separated into three groups based on mean methylation level, blue lines show the top third (high methylation), black lines show the bottom third (low methylation)

BrdU pulse lengths (Additional file 1: Fig. S1 A). We used a probability threshold of $> 0.5$ to call BrdU at each thymidine position on each single-molecule read. With this threshold, we observed a low false positive rate (0.1%) on genomic DNA from cells not treated with BrdU (Fig. 1B), consistent with data from *S. cerevisiae* [39].

Individual BrdU probabilities were used to determine the proportion of thymidine positions substituted with BrdU in windows (of 290 thymidines, corresponding to ~ 1 kb for the human genome) across each single-molecule read (Fig. 1C, D, Additional file 1:

Fig. S1B). Under these conditions, we observed reads with background levels of BrdU incorporation (Fig. 1A,C, left) consistent with parental DNA, and reads with higher BrdU incorporation (Fig. 1A,C, right) consistent with nascent DNA. Analysis of these BrdU substitution levels showed that they have concentration-dependent bimodal distributions (Fig. 1D). For example, with 1.5 or 10 μM BrdU treatments, we observed ∼ 35% or ∼ 65% modal BrdU incorporation, respectively, on the nascent DNA. This indicates that there is a wide range of BrdU pulse concentrations ($\geq 1.5$ μM) that enables parental and nascent DNA to be distinguished. Levels of BrdU substitution are comparable between cell lines and independent of pulse length (Fig. 1D, Additional file 1: Fig. S1B).

We used mass spectrometry to determine the level of BrdU incorporated into bulk cellular DNA for a range of BrdU treatment concentrations. We observe a clear concentration-dependent increase in BrdU incorporation. Next, we used the mass spectrometry data to validate the nanopore sequencing measurements of BrdU incorporation. We see a strong concordance between the two methods across a range of BrdU treatment concentrations, pulse times, and cell lines (Fig. 1E, Additional file 1: Fig. S1 C). For cells subjected to a 50 μM BrdU treatment, the BrdU incorporation level determined by nanopore sequencing is slightly lower than the level determined by mass spectrometry. This is consistent with the DNAscent algorithm slightly under-calling BrdU in DNA with very high levels of BrdU incorporation (e.g., > 80% in the nascent strand for the 50 μM BrdU treatment; Fig. 1D), as previously reported [39]. We also performed BrdU-immunoprecipitation and short-read sequencing (BrdU-IP-seq) from the same cell cultures (Additional file 2: Table S1, Additional file 1: Fig. S1 C). BrdU-IP-seq data does not determine absolute levels of BrdU incorporation, rather relative levels of incorporation between samples that have been barcoded and pooled prior to immunoprecipitation [11]. Although we observe a concentration-dependent increase in nascent strand pull-down, the BrdU-IP-seq data is not linearly related to the amount of BrdU incorporated as determined by mass spectrometry (or nanopore sequencing; Additional file 1: Fig. S1 C).

With cell cultures labelled for a time equivalent to one cell cycle, we can expect up to 50% labelled, nascent reads. We classified reads with $\geq 5\%$ BrdU as nascent strands and found ∼ 40% of such nascent reads in both HeLa-S3 and hTERT-RPE1 cell cultures (Additional file 3: Table S2). These values are consistent with expectation given that some cells exhibit a longer G1-phase [42] and a fraction of cells within cultures are quiescent. Therefore, we have established conditions under which nanopore sequencing can be used to detect BrdU incorporated into nascent human genomic DNA.

### Independent detection of BrdU and CpG methylation

We sought to determine whether nanopore sequencing could independently detect CpG methylation and BrdU incorporation on the same DNA strand without interference. DNAscent was trained on *S. cerevisiae* genomic DNA, which lacks base methylation, and conversely, nanopore detection of methylation has not previously been undertaken in the presence of BrdU. To examine the potential for interference, we called CpG methylation (using Nanopolish [43]) and BrdU incorporation (using DNAscent) in parallel on sequencing reads from cultures treated with a range of BrdU concentrations (Fig. 1F). As previously reported [43], nanopore sequencing for detection of 5 mC agrees well with

bisulfite sequencing datasets (not shown), unaffected by the presence of BrdU in the reads analyzed. For example, very high levels of BrdU incorporation (~ 60%) in a nascent read do not result in an increase in methylation detection (Fig. 1A, C & F). Furthermore, the example nascent read contains a methylated CpG island (70.41–70.42 Mbp) that has not affected the frequency of BrdU calls (Fig. 1A, C & F).

Next, we sought to quantify genome-wide, any effect of 5 mC on BrdU detection by nanopore sequencing. First, hypo- and hypermethylated CpG islands (CGIs) in HeLa-S3 cells were identified using published bisulfite sequencing data [44]. From the published data, we observe ~ 140-fold more methylation in hyper- compared to hypomethylated CGIs. Then, in these CGI categories, we assessed levels of BrdU detection in nanopore sequence reads from HeLa-S3 cell cultures treated with a range of BrdU concentrations (in 100-bp windows; Fig. 1G and Additional file 1: Fig. S1D). In data from cells not treated with BrdU, we observe a BrdU false positive rate of ~ 0.1% at hypomethylated CGIs (Fig. 1G left panel, black line), consistent with the genomic average. Within a narrow window centered on hypermethylated CGIs, we observe a slight increase in base-resolution BrdU false positives, i.e. on DNA not treated with BrdU, peaking at ~ 5% (Fig. 1G left panel, blue line). Conversely, in reads with a high level of BrdU incorporation (~ 80% on nascent reads; Fig. 1D), we observe a slight dip in base-resolution BrdU detection (indicating false negatives) across hypomethylated CGIs (Fig. 1G, right panel, black line). Across hypermethylated CGIs, we observe no variation in BrdU detection (Fig. 1E, right panel, blue line). Hence, even in the most methylated regions of the genome, we observe only minor, localized effects on BrdU detection. In subsequent analyses, we look at BrdU incorporation in windows (of 290 thymidines) where any minor effect of methylation on BrdU detection will be further reduced by > fivefold. Therefore, we consider that variation in methylation does not adversely affect our ability to detect levels of BrdU incorporation in nanopore sequencing reads.

### Whole genome single-molecule identification of replication initiation sites

To identify temporal patterns of DNA replication on nascent single molecules, we treated asynchronously growing cell cultures with increasing concentrations of BrdU (0 to 12 µM in 0.5 µM increments over 1 h, followed by a further 1 h at 12 µM; Fig. 2A). Then, ultra-long nanopore sequencing reads (half of the data contained in reads longer

(See figure on next page.)
**Fig. 2** Single-molecule detection of DNA replication dynamics on ultra-long nanopore sequencing reads. **A** Schematic of the experimental strategy with sequential additions of BrdU for detection of replication dynamics by DNAscent. **B, C** Example nanopore sequencing reads, aligned to the human genome, from HeLa-S3 cells cultured as described in Fig. 2 A. Black dots indicate BrdU probabilities at each thymidine position and the blue line indicates the fraction BrdU incorporation in independent 290 thymidine windows. Above each example read are shown: replication forks with direction (arrows), initiation (light green bars), and termination sites (dark green bars), all computationally determined from DNAscent data. Ok-seq initiation zones (dark blue bars) and ORM initiation zones (yellow bars) are also shown, from HeLa-S3 datasets. **D** The summary plot (upper panel) shows the mean signal across DNAscent initiation sites (± 100 kb) for proportion GC content and initiation zones from Ok-seq (HeLa-S3), Pu-seq (HCT116), and optical mapping (ORM; HeLa-S3) in blue. Also plotted are random expectations (black line) and 99% confidence intervals (gray band) from 1000 randomizations of DNAscent initiation site coordinates. The lower panel represent the same data as heatmaps (IZs in yellow) ordered by replication timing (blue) where each row is an individual DNAscent replication initiation site

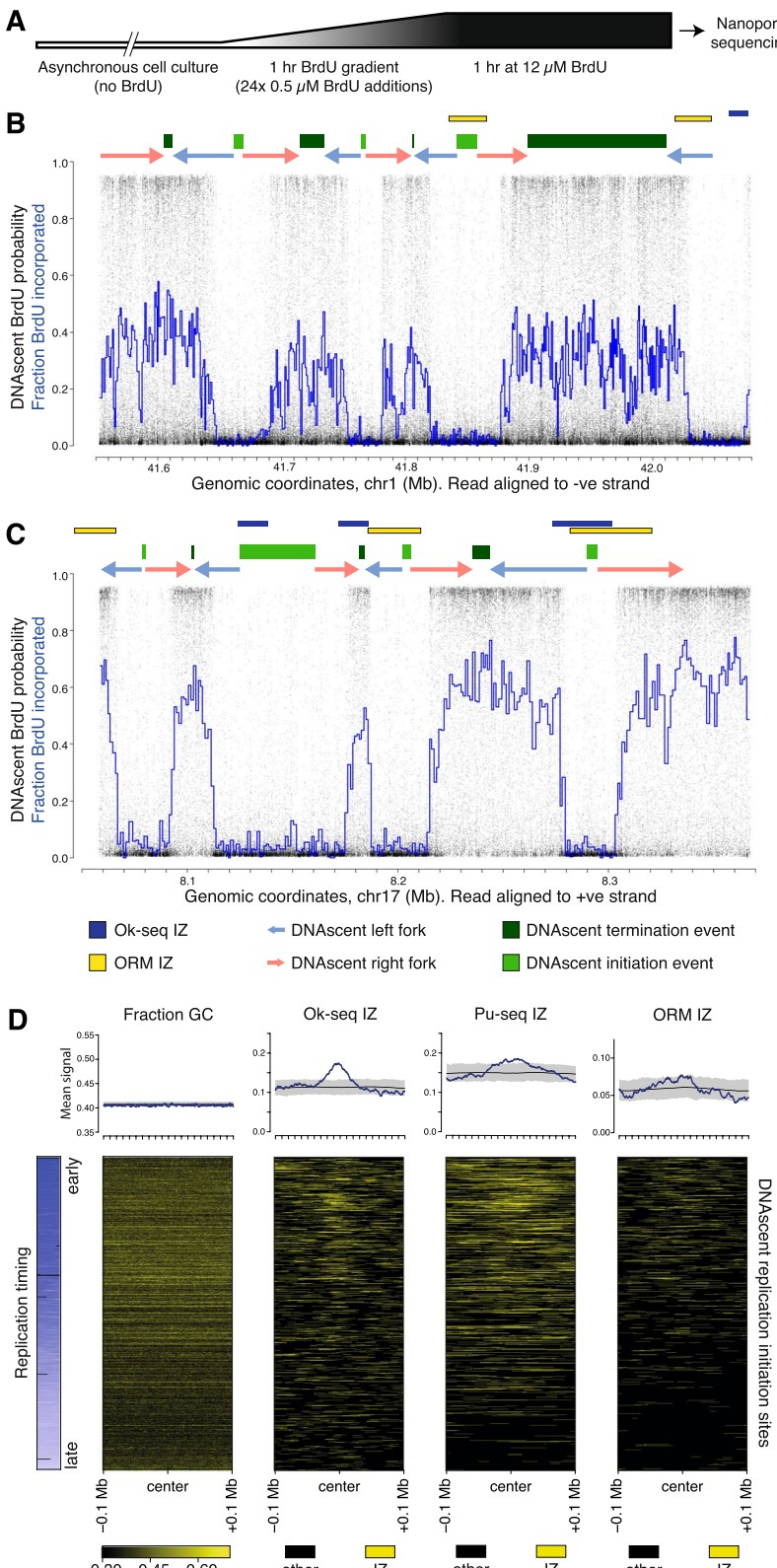

**Fig. 2** (See legend on previous page.)

than (N50) 120 kb; Additional file 4: Table S3 A) were generated and used to determine sites of BrdU incorporation, as described above. This regime of BrdU additions to cells generated regions of increasing BrdU incorporation across sequencing reads from which we could calculate gradients and fork directions. In contrast single pulse treatments of BrdU resulted in rapid transitions from low to high BrdU incorporation (within 1–2 kb) and consequently lower temporal resolution and reduced ability to identify fork direction (data not shown). In Fig. 2B and C, we show two example single-molecule reads visualizing BrdU probabilities at each thymidine (black dots) and windowed levels of BrdU incorporation (blue lines). Background levels of BrdU signal represent sequences replicated prior to the addition and incorporation of BrdU (Fig. 2B, C). Gradients of increasing BrdU incorporation, corresponding to sequences replicated during the sequential BrdU additions, indicate replication fork direction. Therefore, minima and maxima in BrdU incorporation identify DNAscent replication initiation and termination events respectively. On this basis, replication fork direction and DNAscent replication initiation and termination sites were identified computationally genome-wide (see "Methods"; Fig. 2 B, C, fork direction indicated by blue and red arrows, light and dark green bars indicate initiation and termination sites, respectively). We intentionally set a high stringency for initiation site calling to favor specificity (low false positive rate) over some sensitivity (true positive rate), to have high confidence in the initiation sites identified.

For the example 526-kb single-molecule read in Fig. 2B, we identified three DNAscent replication initiation events and four termination events. For the example 309-kb single-molecule read shown (Fig. 2C), we identified four DNAscent replication initiation events and three termination events. We note that the minima (DNAscent replication initiation events) vary in width, for example the minimum flanking 8.15 Mb is wider than the other three minima on this molecule (Fig. 2C). We consider two explanations. First, wide minima could result from multiple initiation events with replicons merging prior to the BrdU addition. Second, the width of minima could indicate how far replication forks have progressed from a single initiation event prior to BrdU addition. In the second scenario, the minima width is a measure of relative replication initiation time across a molecule, with wider minima arising from earlier replication initiation events. Given reported distances between replication initiation events of ~ 100 kb [33–35], we favor this second explanation for these molecules, especially for narrower minima. Analogous to the scenarios at minima, wider maxima could either arise from a single replication termination event or multiple events occurring during the final 1 h at 12 µM BrdU where we do not have temporal resolution.

At the depth of sequencing performed for the HeLa-S3 sample, we identified a total of 2577 DNAscent replication initiation sites (Additional file 4: Table S3B, these sites were used for subsequent initiation site density analyses) and 2791 termination sites. For hTERT-RPE, we find 912 initiation sites and 1099 termination sites. In both datasets, BrdU containing reads and identified replication initiation sites are distributed across the genome (Additional file 1: Fig. S2 A & B). Next, we filtered the set of initiation sites to those with a resolution of < 5 kb (dashed line in Additional file 1: Fig. S2 C), which are most likely to result from a single initiation event. This identified 1690 high-resolution HeLa-S3 DNAscent initiation sites that were used for all intersection analyses. Comparisons to published population-level relative replication timing data (sort-seq from

the same HeLa-S3 stock [13]) show that we have identified initiation sites throughout S phase (Fig. 2D left panel). These observations are consistent with expectations, given that labelling was performed on asynchronously growing cell cultures with an unbiased representation of all stages of S phase.

To compare the density of DNAscent replication initiation sites between different genomic regions, we determined the number of *r*eplication *i*nitiations per *g*igabase of mapped *r*eads (abbreviated to RIGR). Across four equally sized replication timing quartiles, we observed a slight enrichment in replication initiation density within the earliest quartile (15% more than expectation; $p < 0.00005$; Additional file 5: Table S4). We also observe a decrease in GC-content for DNAscent initiation sites from later replicating regions of the genome (Fig. 2D, panel 2). This may be a consequence of lower gene density in later replicating parts of the genome [45]. However, we do not observe any localized variations in DNA sequence composition or enriched sequence motifs associated with DNAscent initiation sites (see "Methods"), consistent with low sequence specificity for human ORC [8].

### Initiation sites identified by single-molecule sequencing are enriched in Ok-seq initiation zones

The 1690 high-resolution DNAscent replication initiation sites were compared with initiation sites and zones reported by other studies (in the same cell line where possible). For the example reads described above, of the seven DNAscent initiation sites, three intersect with published HeLa-S3 ORM initiation zones (Fig. 2 B, C; yellow bars) [37] and two intersect with published HeLa-S3 Ok-seq initiation zones (Fig. 2B, C; blue bars) [16]. Across the high-resolution DNAscent initiation sites identified in HeLa-S3 cells, we observe a clear enrichment in published ORM [37], Pu-seq [17], and Ok-seq [16] initiation zones, that is most pronounced in early S phase (Fig. 2D). We determined the relative distance between Ok-seq initiation zones and DNAscent initiation sites to test for spatial correlation. DNAscent initiation sites and Ok-seq initiation zones occur with much closer proximity than expected by random chance ($p < 0.001$; Additional file 1: Fig. S2D).

Given the strong enrichment of published Ok-seq initiation zones within and in close proximity to DNAscent initiation sites (Fig. 2D; Additional file 1: Fig. S2D), we next examined the reciprocal relationship. In HeLa-S3 cells, Ok-seq previously identified 8984 replication IZs with a mean size of 32 kb [16]. Within these Ok-seq IZs, we identified 507 DNAscent replication initiation sites in HeLa-S3 cells. For example, on chromosome 15, a 599-kb single-molecule sequencing read identified two DNAscent replication initiation sites both of which are contained within Ok-seq IZs (Fig. 3A). Overall, within Ok-seq IZs, we find a DNAscent initiation site density that is double expectation (RIGR $= 157$ compared to an expectation of 77.8 from simulations; $p < 0.00001$; Additional file 5: Table S4)—we term these "focused" DNAscent initiation sites. The high density of DNAscent initiation sites in Ok-seq IZs is observed across S phase but is greatest in the earliest (first) replication timing quartile (RIGR $= 180$) and progressively falls through the second (RIGR $= 155$), third (RIGR $= 124$) and fourth (RIGR $= 104$) timing quartiles. In summary, DNAscent identifies high-resolution replication initiation sites on single

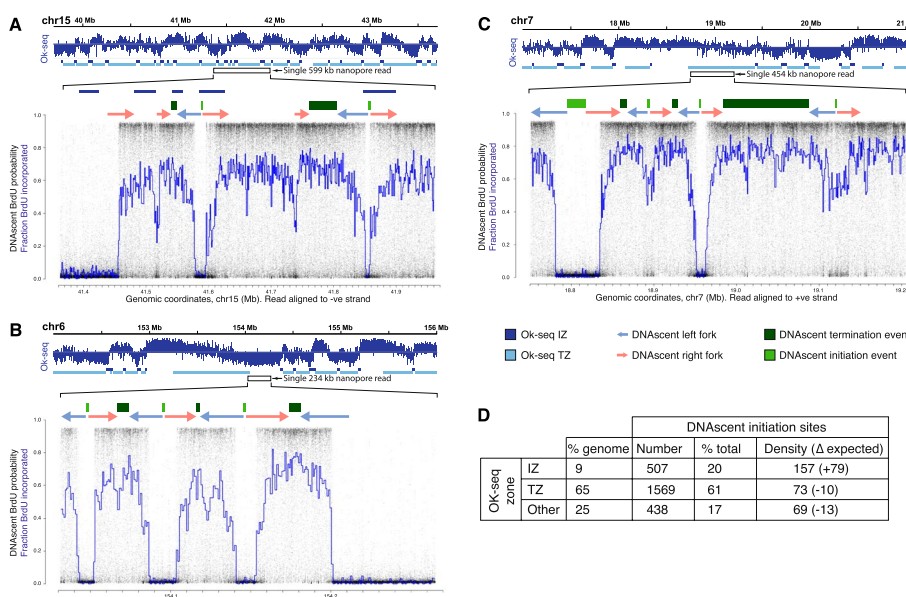

**Fig. 3** DNAscent identifies dispersed replication initiation sites not found by population studies. **A–C** Upper panels each show 4 Mb of published strand-specific population-level Okazaki fragment sequencing depth and the determined Ok-seq initiation (dark blue) and termination zones (light blue) [16]. Lower, zoomed in panels show ultra-long single-molecule nanopore sequence reads with DNAscent BrdU probabilities (black dots) and fractions BrdU incorporated (blue lines). As in Fig. 2B, above each read are shown DNAscent replication forks with direction (arrows), initiation (light green bars), and termination sites (dark green bars). **A** Example read showing two DNAscent initiation sites corresponding with published Ok-seq initiation zones. **B** Example read showing three DNAscent initiation sites within a region determined from population average Ok-seq data to be predominantly replicated by leftward forks and absent of IZs. **C** Example read showing four DNAscent initiation sites within a region determined from population average Ok-seq data to be a termination zone. **D** Number, proportion, and density (RIGR) of DNAscent initiation sites found in Ok-seq initiation and termination zones as well as regions classed as other (including predominantly unidirectional forks). Also shown is the percentage of the genome that these classes occupy. Difference (Δ) from expectation was determined from 1000 randomizations of DNAscent initiation site coordinates

molecules, throughout S phase, that are enriched in the initiation zones identified by published population-level replication dynamics studies.

## Most replication initiation occurs outside of initiation zones

Although we observe a strong enrichment of DNAscent initiation sites within and in close proximity to published Ok-seq initiation zones, we note that this was the case for only a subset of sites (Additional file 1: Fig. S3 A & B). For example, despite this significant enrichment, only 20% of the DNAscent initiation sites lie within Ok-seq IZs (focused sites; 31% in the first quartile of S phase, falling to just 5% in the fourth quartile of S phase; Additional file 5: Table S4). Therefore, we undertook a more detailed comparison between these datasets. In Ok-seq data, the proportion of reads mapping to each strand serves as a proxy for replication fork direction (RFD). Three features have previously been described from genomic plots of Ok-seq RFD: sharp positive gradients identifying IZs (Fig. 3A–C, upper panels, indicated by dark blue boxes), plateaus potentially consistent with a single progressing replication fork (Fig. 3B, C, upper panels, indicated by the absence of blue boxes), and gradual negative gradients identified as replication

termination zones (TZs; Fig. 3A–C, upper panels, indicated by light blue boxes). We identified DNAscent replication initiation sites across all three Ok-seq features.

DNAscent initiation site density (RIGR) is ~50% lower outside of Ok-seq IZs and lower than expected by a random distribution (Fig. 3D). However, Ok-seq plateaus and TZs cover ~2 × and ~7 × more of the genome respectively than Ok-seq IZs. Therefore, despite the lower initiation density, 19 and 61% of all DNAscent initiation sites intersect with Ok-seq plateaus and "termination" zones, respectively—we term these "dispersed" DNAscent initiation sites. We see clear examples of multiple replication initiation sites on individual molecules that span plateaus of population-level RFD (e.g., Fig. 3B) and within regions designated by population-level data as termination zones (e.g., Fig. 3C).

When considering all high-resolution (< 5 kb) DNAscent initiation sites, we observed significant enrichment with Ok-seq, Pu-seq, and ORM IZs (Fig. 2D), but not with the published sites identified by chromatin immunoprecipitation (ChIP; ORC [26] or Mcm7 [27]), SNS-seq [46], or Ini-seq [23, 24] (Additional file 1: Fig. S3 A, C). However, when we consider just focused DNAscent initiation sites, we see a modest significant enrichment for Mcm7-ChIP [27], Ini-seq [23, 24], and SNS-seq sites [24, 46] (Additional file 1: Fig. S3B, D). Therefore, ~20% of DNAscent initiation sites showed enrichment for the published initiation sites identified by a range of independent population-based genomic assays. However, the majority (80%) of replication initiation sites are missed by all population averaged datasets and are dispersed throughout the genome.

### Targeted single-molecule sequencing identifies replication initiation upstream of the TOP1 gene

The coverage of nascent DNA reads span the genome, but multiple nascent reads at the same genomic loci are rare (Additional file 1: Fig. S2 A & B). Therefore, to generate higher coverage at selected loci of interest, we combined sequential BrdU addition (as above) with a molecular target enrichment protocol, nanopore Cas9-targeted sequencing (nCATS) [47]. We expect to find a high frequency of replication initiation events occurring within a narrow genomic window at focused initiation sites, and elsewhere lower frequency of replication initiation events in regions of dispersed initiation. To test this, we designed guide RNA sequences to enrich four ~50-kb genomic regions (Additional file 4: Table S3 C) previously reported to include replication initiation sites in proximity to the TOP1, MYC, AFF2, and HBB genes [48–51]. Based upon our above definitions, the TOP1 and MYC loci intersect with Ok-seq IZs and therefore initiation sites at these loci we class as focused (Fig. 4A & Additional file 1: Fig. S4 A). In contrast, AFF2 and HBB loci intersect with an Ok-seq TZ and plateau, respectively, and therefore initiation sites within these loci we class as dispersed (Additional file 1: Fig. S4B & C). At the AFF2 and HBB loci, we anticipate more dispersed initiation that may rarely lie within the target enrichment region. Overall, we observe strong enrichment with the nCATS protocol, with ~90, ~180, ~190, and ~295-fold enrichment at AFF2, TOP1, HBB, and MYC gene loci, respectively (Additional file 4: Table S3D). Resulting nanopore sequence data were analyzed as described above to determine the patterns of BrdU incorporation across individual single molecules, computationally identifying replication forks, initiation sites, and termination sites. Individual nascent reads with at least one called

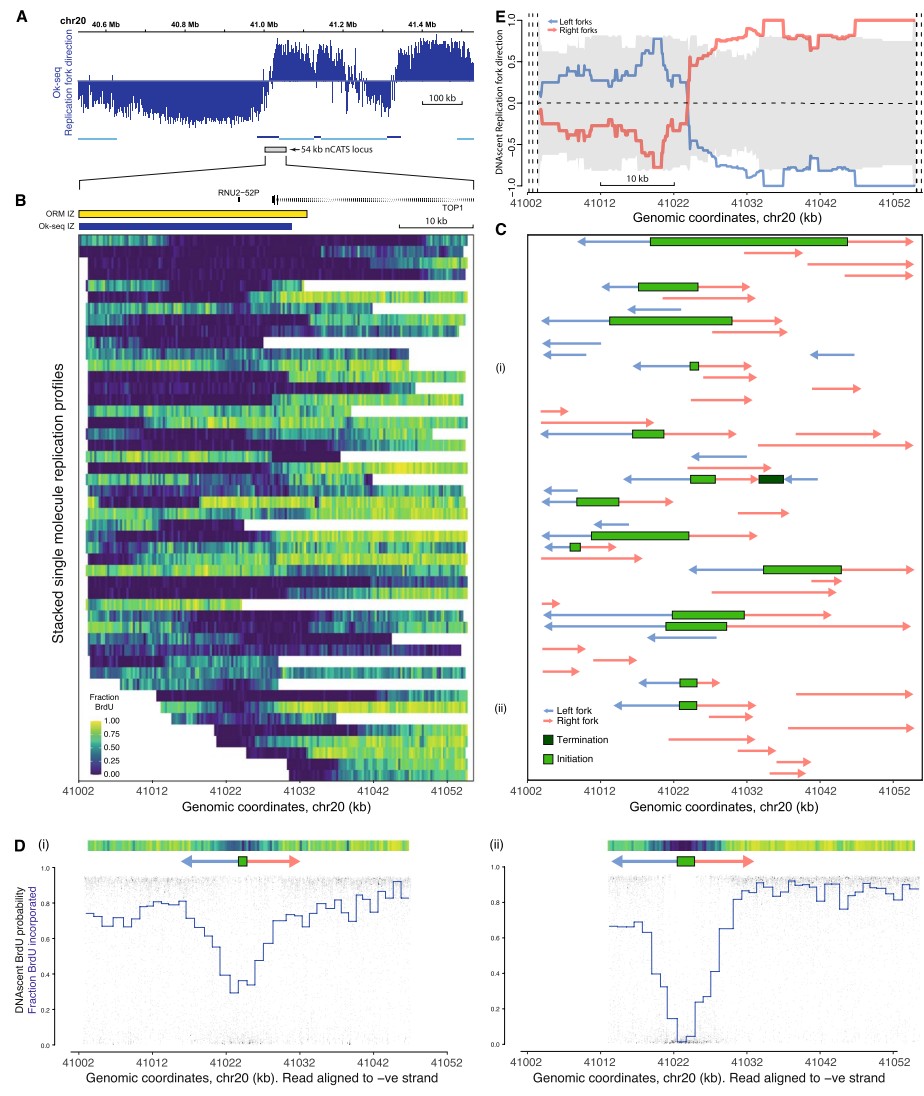

**Fig. 4** Focused replication initiation upstream of the TOP1 transcription start site. **A** Replication fork direction from published Okazaki fragment sequencing [16] across 1 Mb of the human genome centered on the TOP1 gene transcription start site. The location of Ok-seq determined initiation (dark blue) and termination zones (light blue) are marked. Gray box indicates the 54-kb locus targeted for enrichment via nCATS. **B** Stacked single-molecule DNAscent DNA replication profiles with fraction BrdU incorporation indicated via the heatmap (300-bp windows). Gene annotations and the location of ORM and Ok-seq IZs are marked above the heatmap. **C** For each single-molecule DNAscent DNA replication profile, computationally determined replication kinetics are shown: forks with direction (arrows), initiation (light green bars) and termination sites (dark green bars). **D** Two example single-molecule nanopore sequence reads are plotted with DNAscent BrdU probabilities (black dots) and fractions BrdU incorporated (blue lines) as in Fig. 2. For each read, the respective heatmap and replication kinetics are reproduced from panels **B** and **C**. **E** Ensemble of DNAscent single-molecule replication fork direction across the target enrichment locus. The gray ribbon indicates the 95% confidence limits from 1000 randomizations of individual replication fork directions

replication fork are shown, with BrdU incorporation represented by heatmaps (Fig. 4B & Additional file 1: Fig. S4).

At TOP1 and MYC, we achieved nascent strand coverage of 125 and 242 reads and identified 14 and 19 initiation events respectively (RIGR = 103.2 and 84.8, Additional file 4: Table S3D). At TOP1, the majority of DNAscent initiation events are upstream

of the TOP1 gene. We identified six high resolution (mapped to < 5 kb) replication initiation events within a region of chromosome 20 from 41,008 to 41,028 kb (Fig. 4C). This region overlaps with Ok-seq and ORM IZs. For example, two independent single molecules identified replication initiation events at 41,024–41,025 kb and 41,023–41,025 kb (Fig. 4D). At MYC, the majority of identified DNAscent replication initiation events are downstream of the MYC gene (14 high resolution events mapped within a region of chromosome 8 from 127,745 to 127,764 kb) and overlap with the Ok-seq and ORM IZs (Additional file 1: Fig. S4 A). For example, on two independent single molecules, we identify replication initiation events at 127,758–127,760 kb and 127,753–127,755 kb (Additional file 1: Fig. S4 A). While these two initiation events are close, they do not overlap, consistent with multiple proximal initiation sites within this initiation zone.

At the predicted dispersed sites AFF2 and HBB, we generated nascent coverage of 29 and 99 reads and identified one and three replication initiation events respectively (RIGR = 16.4 and 19.6; Additional file 1: Fig. S4; Additional file 4: Table S3D). This is consistent with our expectation of reduced initiation site density at dispersed sites (see "Discussion"). Overall, target enrichment combined with DNAscent allows the identification of replication initiation events on multiple independent single molecules at selected loci.

### Focused initiation sites show clear switches in fork direction

Ensembles of fork direction from single molecules across each locus should resemble cell population fork direction data (i.e., Ok-seq or Pu-seq). In Ok-seq data, replication fork direction (RFD) can be calculated as the difference between the number of reads mapping to each strand divided by the total number of mapping reads [52]. Therefore, RFD ranges from -1 (100% leftward forks) to +1 (100% rightward forks) and 0 indicates an equal proportion of leftward and rightward forks. At TOP1 and MYC, ensembles of our single-molecule data show a clear switch in predominant fork direction at the most frequently identified sites of replication initiation consistent with cell population-level Ok-seq data (Fig. 4E & Additional file 1: Fig. S4 A). However, across the targeted dispersed initiation site loci, we do not observe a clear switch in predominant fork direction. At HBB, the ensemble of single-molecule fork direction data indicates a predominant leftward direction (31 leftward and 18 rightward forks; RFD = − 0.27, Additional file 1: Fig. S4B) consistent with Ok-seq data (mean RFD value of − 0.43). At AFF2, we only identified a total of 11 gradients with similar numbers of leftward (5) and rightward (6) forks (RFD = 0.09; Additional file 1: Fig. S4 C) again consistent with Ok-seq data (mean value of 0.10). Taken together, our high coverage single-molecule data across selected target enrichment loci allows high-resolution determination of replication fork direction and initiation site identification.

The increased resolution of our single-molecule data over published Ok-seq data refines the sites of replication initiation and allows determination of the predominant direction of replication across neighboring genes. Interestingly, across the start of the highly expressed long (~ 96 kb) TOP1 gene, we observe that there is predominantly codirectional replication and transcription. However, across the highly expressed short (~ 7.6 kb) MYC gene, we predominantly observe head-on replication and transcription.

**Transcription excludes replication initiation promoting co-directionality**

For expressed genes, population-level studies of replication dynamics (e.g., Ok-seq) report a bias for codirectional replication and transcription at transcription start sites (TSSs); and a counter-directional fork bias at transcription end sites (TESs) [53]. As described above, the majority of replication initiation events detected on single molecules are missed by population-level data (Fig. 3). Related to this, we observe many replication forks moving in a direction opposite to the average reported by bulk population data (e.g., rightward forks in Fig. 3B). Furthermore, two of the loci we selected for target enrichment include highly expressed genes and we observed predominantly codirectional replication and transcription at TOP1, but a counter-directional bias at MYC. Therefore, we tested whether our genome-wide single-molecule replication fork directions displayed a bias at highly transcribed genes. Transcribed genes were divided into two categories; "low" and "high" transcription (based upon nucleoplasmic RNA-seq from HeLa-S3 cells [54]). The median gene transcription is ~30-fold greater in the high transcription compared to the low transcription category (each containing 8123 genes). At genes with a low level of transcription, there is no significant bias in replication fork co-directionality (Fig. 5A, left panel). At genes with a high level of transcription, co-directional replication forks are significantly overrepresented from 3 kbp upstream of the TSSs and at least 10 kbp into the gene body (Fig. 5A, right panel, green line; $p < 0.01$), with long genes (> 42.5 kbp) accounting for most of the effect (Additional file 1: Fig. S5 A). Counter-directional replication and transcription is significantly underrepresented across the same region of highly transcribed genes (Fig. 5A, right panel, blue line). We observe similar replication fork biases at TSSs in the RPE1 dataset (Additional file 1: Fig. S5B). Additionally, we observe a modest trend towards overrepresentation of counter-direction replication upstream of the TESs of highly transcribed genes, although no individual point passes a 95% confidence threshold (Additional file 1: Fig. S5 C & D).

The observations above suggest that replication initiation may preferentially take place flanking highly transcribed genes. As a direct test, we determined the density of DNAscent replication initiation sites (RIGR) within and flanking genes with low and high levels of transcription (as defined above). Across genes with a low level of transcription, we see no significant variation in DNAscent initiation site density compared to random expectations (Fig. 5B, left panel). However, in highly transcribed genes, we see a significantly lower density of DNAscent initiation sites within the transcribed portion (RIGR $= 50.3$, compared to a mean of 81.8 from randomizations; $p < 0.00001$; Fig. 5B, right panel). In contrast, DNAscent replication initiation site density is elevated within a 25-kb window upstream of the highly transcribed genes (RIGR $= 126.2$, compared to a mean of 73.7 from randomizations respectively; $p < 0.00001$; Fig. 5B, right panel). Therefore, we observe that highly transcribed genes tend to exclude replication initiation from the gene body, with initiation often taking place in upstream regions.

Next, we tested whether proximity to high transcription is a general property of DNAscent replication initiation sites or is specific to the focused subset. Figure 5C visualizes transcript abundance within 100 kb of each DNAscent replication initiation site (Fig. 5C left heatmap early S phase; Additional file 1: Fig. S5E heatmap all sites; other transcription-associated chromatin marks, Additional file 1: Fig. S5E). We separately considered focused and dispersed DNAscent replication initiation

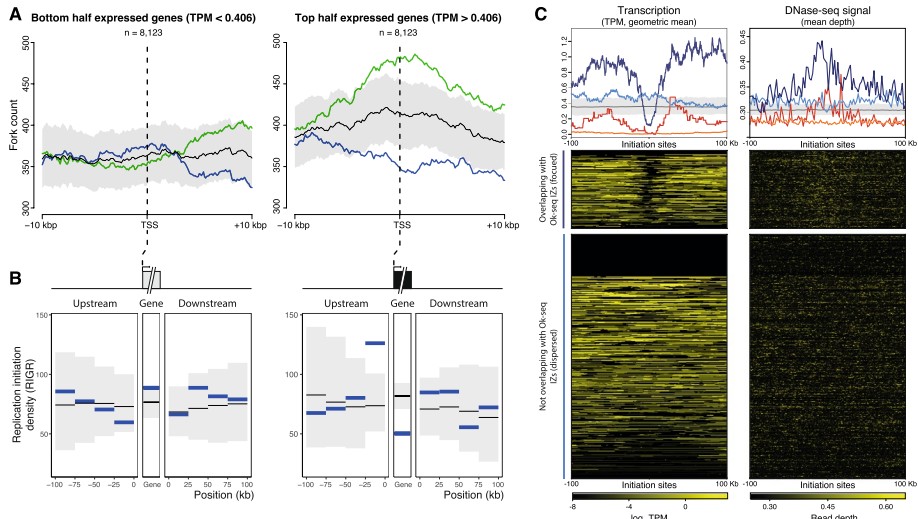

**Fig. 5** A fifth of initiation sites are focused by high levels of proximal transcription to define initiation zones. **A** Replication fork direction relative to the level and direction of transcription. Genes were categorized as low (left) or high (right) transcription based on ranked normalized transcript counts (Transcripts Per Million; TPM). Replication forks, identified by DNAscent, occurring within 10 kb of transcription start sites were included. The count of replication forks co-directional with transcription is shown in green and counter-directional in blue. Confidence intervals (99%) from 1000 fork direction randomizations are shown in gray around average fork count (in black). **B** Replication initiation is reduced in highly transcribed genes. Blue lines indicate normalized DNAscent initiation site density (RIGR) within and adjacent (100 kb up- and downstream) to the bottom (left) or top half (right) of transcribed genes (as in Fig. 5 A). Thin black lines show average density from 1000 initiation site randomizations with 99% confidence intervals (gray ribbon). **C** Normalized transcription counts (TPM; left) and DNase-sensitivity (read-depth; right) 100 kb up- and downstream of DNAscent replication initiation sites separated by intersection with Ok-seq IZs and replication time. (Upper panel) Geometric Mean transcription counts (TPM; left) and DNase-seq signal (mean read depth; right) for each set of DNAscent replication initiation sites (dark blue is early S phase focused sites, $n = 218$; light blue is early S phase dispersed sites, $n = 681$; red is late S phase focused sites, $n = 66$; orange is late S phase dispersed sites, $n = 716$). Black line is the mean from 1000 randomizations of DNAscent initiation site locations, and the gray ribbon covers the 99% confidence interval. (Lower panel) Heatmaps for individual early S phase focused or dispersed DNAscent replication initiation sites. The heatmap key indicates $\log_2$ TPM (left) and DNase-seq read depth (right)

sites with each subdivided by replication time. We observe clear minima in transcription centered on the focused DNAscent replication initiation sites (dark blue and red plots). This is clearest in early S-phase (dark blue), but apparent for the smaller number found within late S-phase (red). We see no evidence for a similar relationship to transcription for the more numerous dispersed DNAscent initiation sites (light blue and orange plots). Furthermore, the focused DNAscent initiation sites are also enriched for multiple signals of accessible/open chromatin (e.g., DNase-seq read depth, Fig. 5C right heatmap early S phase; Additional file 1: Fig. S5E), unlike the dispersed DNAscent initiation sites. Therefore, our single-molecule replication dynamics data demonstrate a clear difference in transcriptional and epigenetic context between replication initiation sites that are focused in IZs and the majority that are dispersed throughout the genome. In summary, we report a novel class of "dispersed" replication initiation sites, undetectable by population-level studies, that replicate most of the genome.

## Discussion

For the first time, we report DNA replication initiation sites identified genome-wide in human cell cultures using ultra-long single-molecule sequencing. To achieve this, we have established and validated experimental conditions for concentration-dependent BrdU incorporation and quantitative detection with nanopore sequencing (Fig. 1). We show that there is minimal interference between Me-CpG and BrdU detection in nanopore sequencing data, allowing independent quantification of both base modifications (Fig. 1F, G). Sequential additions of BrdU to the culture media generated gradients of BrdU incorporation on resulting single-molecule sequencing reads (Fig. 2A–C). This enabled replication fork direction to be determined and sites of initiation and termination to be discovered. By combining DNAscent with nCATS target enrichment, we generated higher coverage for selection regions of interest, identifying multiple overlapping replication initiation events. This method is performed in unperturbed cells and uses low levels of analog in the culture media thereby avoiding the risk of inducing DNA damage and/or perturbing replication fork dynamics. Additionally, the fork direction information is generated without the need for high concentrations of a second potentially more toxic analog [55–57] as used in double labelling protocols. Therefore, we present a method to identify DNA replication initiation and termination sites in human cells that is readily transferable between cell types and to answer various biological questions.

Using ultra-long single-molecule sequencing, we have identified and located thousands of high-resolution DNA replication initiation and termination events in cultured human cells. These include reads that feature multiple replication initiation and termination events (Figs. 2 and 3). The use of unperturbed, asynchronous, cells has allowed us to identify replication initiation events throughout S-phase (Fig. 2D). These are functionally activated origins that we refer to as replication initiation sites. Given the high numbers of MCM double-hexamers loaded onto DNA in human cells [58, 59], these initiation events likely reflect only a fraction of licensed sites. Due to sequencing depth limitations, our whole genome single-molecule dataset samples initiation sites rather than exhaustively identifying all sites. When enriching for selected loci, we achieved nascent strand coverage of up to 242 and identified up to 19 single-molecule replication initiation events within a 50-kb window. Overall, the frequency of initiation sites that we identify are consistent with a model that many more sites are licensed (with MCM) and identified by methods such as MCM ChIP-seq, than are functionally activated to initiate replication forks.

A lack of concordance between various published methods for identifying DNA replication initiation sites has frequently been reported [10, 28, 37]. Here we identify replication initiation sites that are enriched in the initiation zones identified by population-level replication dynamics studies (Ok-seq, PU-seq) and initiation zones from an optical mapping study (ORM) (Fig. 2D, E) [16, 17, 37]. Remarkably, when considering DNAscent focused initiation sites, we also find concordance with other major published cell population methods (Additional file 1: Fig. S3B), despite previous comparisons between these methods not showing concordance. We propose that focused sites are used at higher frequency and therefore identifiable by cell population methods. By filtering these cell population datasets by the focused initiation sites identified here, the comparisons are less susceptible to method-specific noise. However, we do not find any significant sequence

motif enrichment in the initiation sites we identified, whether considering all or only the focused initiation sites.

To confirm that focused replication initiation sites are activated at high frequency and consistently used within the cell population, we used targeted enrichment (nCATS) to selectively sequence two ~50-kb regions intersecting with focused initiation sites (MYC and TOP1). At both sites the single-molecule sequencing data identify many replication initiation events with remarkable consistency in initiation sites between individual cells (Fig. 4 and Additional file 1: Fig. S4 A). By contrast, we observe a far lower density of replication initiation events at two sites (AFF2 and HBB) that we define as dispersed. This was anticipated: the size of each targeted site was ~50 kb, but replication initiation sites are only likely to be identified in the central ~30 kb, due to the requirement for detection of flanking bi-directional replication forks. Hence, with an average distance between replication initiation events of ~100 kb and in the absence of a focused initiation site, we would anticipate dispersed initiations within the central 30-kb window to be approximately 30% of the level observed at focused sites. This estimate is similar to the difference in replication initiation density that we observe (RIGR at MYC and TOP1 of 89 and 103 respectively, compared to at AFF2 and HBB of 16 and 20 respectively, i.e., the two dispersed sites are ~20% of the level observed at the two focused sites). Therefore, target enrichment confirms our findings from the whole genome dataset and demonstrate that at both TOP1 and MYC there is remarkable consistency in the location of replication initiation.

Although we show enrichment of replication initiation sites in the initiation zones identified in cell populations, this accounts for only 20% of DNAscent initiation sites with 80% situated outside of IZs (Fig. 6). We find that within the replication IZs identified by population-level studies, there is a much higher density of DNAscent initiation sites (higher RIGR) compared to a lower density of initiation sites across the rest of the genome (Fig. 3D). Notably, we observe initiation sites within regions marked as termination zones in population-level data (Fig. 3C) and termination sites within population-level IZs (Fig. 3A). While large regions of the genome appear unidirectionally replicated in cell population level Ok-seq data, we identify initiation events within these regions. Given that these regions are often much larger than the average IOD of ~100 kb, this is perhaps unsurprising. We suggest that at a single cell/molecule level initiation occurs within these regions at the expected average of once per ~100 kb but that these sites are dispersed such that when analyzing cell population data, the various initiation events are "averaged out" with little to no preference for initiation at particular sites (i.e., no IZ or focussed initiation sites). Our single-molecule data, with multiple independent BrdU incorporation measurements across each read, allows high-confidence identification of individual replication initiation events, even when the site may be rarely used within a population of cells. Therefore, this method is well-suited to identifying the numerous initiation sites, from which the majority of the genome is replicated, but are individually rarely used and therefore missed by population datasets.

We propose that there are genomic regions that strongly favor replication initiation. In these regions, replication initiates in a sufficiently high proportion of cells to permit detection by cell population studies (Fig. 6). However, most replication initiation sites are more spatially dispersed with high cell-to-cell variability thus preventing

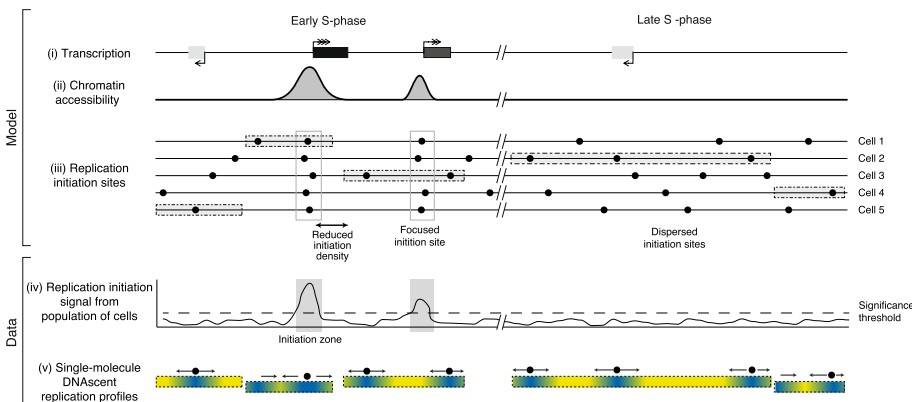

**Fig. 6** Model of the major determinants of human replication initiation site distribution. Two example genomic locations are shown, one replicating in early S-phase (left) and the other in late S-phase (right). (panels *i–iii*) A model to illustrate how the transcriptional (*i*) and chromatin (*ii*) landscape may influence sites and zones of DNA replication initiation (*iii*). (panels *iv–v*) Illustration of the types of replication initiation events that can be detected by population versus single-molecule genomic assays. In panel (*i*), the rectangles indicate high (black with triple arrowhead), medium (gray with double arrowhead), and low (light gray with single arrowhead) levels of transcription. In panel (*ii*) the peaks indicate regions of accessible chromatin, for example the signal from DNase I hypersensitive site sequencing. Panel (*iii*) illustrates sites of replication initiation (black circles) in a population of 5 example cells. Note, dormant origins are not marked. Open boxes illustrate zones of focused initiation sites within which multiple cells within the population initiate DNA replication. Gray-shaded rectangles indicate genome fragments detected by single-molecule sequencing in panel (*v*). Panel (*iv*) illustrates the nature of replication initiation signal from a population of cells (for example, Ok-seq), with gray-shaded rectangles highlighting two initiation zones. The dashed horizontal line illustrates the significance threshold for identifying initiation zones. Panel (*v*) illustrates five example single-molecule DNAscent replication profiles from the example cells in panel (*iii*). In each molecule, the white-black gradient illustrates low–high BrdU incorporation with replication fork direction indicated by arrows. Single-molecule replication initiation events are visualized by filled circles between diverging arrows with equal sensitivity to detect focused and dispersed initiation events in early and late S phase

their detection in cell population-level analyses. We suggest this is due to the probability of these sites firing within each cell, and therefore across a population, being comparatively low. We do not identify any common chromatin feature (from population data) between dispersed initiation sites, but it remains possible that such a signature is present in the single cell that a particular dispersed initiation event took place. Nevertheless, we cannot exclude the possibility that these dispersed sites are largely random. Overall, our model can reconcile the order-of-magnitude difference between the spacing of population-level IZs (megabase) and single-molecule inter-origin distances (IODs; ~ 100 kb). In addition, Ok-seq and ensembles of single-cell DNA replication sequencing (scRepli-seq) [30] imply megabase regions of unidirectional fork movement [16]. However, our data clearly identify multiple dispersed replication initiation sites across such regions. The paucity of initiation sites within highly transcribed regions, observed in our data (Fig. 5B) and from population-level Ok-seq studies [16, 53], leads us to propose that transcription may be one mechanism that strongly determines regions of favored initiation. This could be due to a permissive chromatin environment (Fig. 5C, Additional file 1: Fig. S5E) adjacent to active genes and/or by displacing MCMs from gene bodies [59] (Fig. 6). Therefore, we propose that high levels of transcription confine replication initiation to consistent sites within the cell population, which we term "focused" sites, that are thus observable

by cell-population techniques (Fig. 6). In our dataset, 20% of replication initiation events are located within initiation zones—i.e., focused sites; whereas the other 80% of initiation events are located at sites dispersed throughout the genome (including within transcribed genes; Fig. 5C), with each site being used at low frequency within the population (Fig. 6).

Our demonstration of DNAscent in cultured human cells opens the way for numerous future studies with the potential to address further longstanding biological questions. For example, DNAscent can be applied to look at the roles of cis- and trans-acting factors in DNA replication and models of human disease via perturbation of gene function. DNAscent is applicable to a wide range of cell types and different organisms—requiring only that cells can be cultured, that cells incorporate BrdU, and that high molecular-weight DNA can be extracted. Ultra-long molecules combined with improved reference genomes, for example recent telomere-to-telomere assemblies, will allow analyses of DNA replication in understudied repetitive regions of genomes, including centromeres and telomeres. Furthermore, gradients of analog incorporation have been shown to allow detection of replication fork pause sites [38]. Variants of the BrdU incorporation regime [60] or the use of multiple analogs [61] allows quantification of replication fork kinetics to identify the genomic context of challenges to replication fork progression. Additionally, independent detection of DNA methylation and BrdU incorporation on the same molecules will enable assessment of how methylation status impacts DNA replication dynamics and the kinetics of methylation re-establishment on nascent DNA.

More broadly, in addition to our single-molecule detection of DNA replication dynamics, we envisage that DNAscent will allow detection of BrdU incorporated through other cellular pathways, including DNA repair synthesis. Moreover, DNAscent could be used as a molecular tool to detect ex vivo labelled single-stranded DNA breaks, either those induced in vivo or generated in vitro, for example with a DNA glycosylase. Furthermore, steadily increasing nanopore sequencing throughput will allow the identification of more BrdU-labelled sites allowing greater statistical power in subsequent analyses.

Optical methods that visualize the incorporation of modified nucleotides into DNA have provided enormous biological insights into pathways of DNA replication, recombination, and repair. However, to date these methods have been limited by low throughput, low spatial resolution, a lack of the underlying sequence context and/or the requirement for perturbation to S phase progression. Our application of nanopore sequencing provides a step change to these powerful methods, to provide quantitative, high-throughput, high-resolution, sequence-specified detection of base analogs. This has allowed, for the first time, the discovery of replication initiation sites on single sequence-resolved molecules across the human genome.

## Conclusions

Using ultra-long single-molecule nanopore sequencing to detect BrdU, we demonstrate unbiased and highly sensitive detection and characterization of DNA replication initiation and termination events in human cells. We discovered that the majority of the human genome is replicated from dispersed initiation sites that are individually rarely used and therefore missed by population datasets.

## Methods

### Cell line maintenance and BrdU treatment

HeLa-S3 (adherent) and hTERT-RPE1 were obtained from the American Type Culture Collection (ATCC) and maintained in DMEM Glutamax (HeLa-S3) or DMEM/F12 Glutamax (hTERT-RPE1, both Gibco), with the addition of 10% fetal bovine serum (Sigma) and 1% penicillin/streptomycin (Gibco). Cells were maintained at 70% confluency in 5% $CO_2$. All the cell lines were tested negative for mycoplasma.

For BrdU pulse concentration scoping experiments, asynchronous cultures were treated with the indicated concentrations (0.3–50 μM) of BrdU for 20 h (HeLa-S3) or 2, 24, or 27 h (hTERT-RPE1). For BrdU gradient experiments, asynchronous HeLa-S3 or hTERT-RPE1 cell cultures were treated with BrdU from 0 to 12 μM over 1 h, with the addition of 0.5 μM every 2.5 min until 12 μM which was incubated for a further 1 h.

### Sample harvesting

At appropriate time points, samples were harvested for genomic DNA for long-read nanopore sequencing, BrdU-IP short-read sequencing, and mass spectrometry. Cells were washed twice in ice-cold D-PBS and scrape harvested. Pellets were collected by centrifuging samples at 500 $g$ for 5 min at 4 °C and stored at -20 °C.

### Standard DNA extraction

For genomic DNA extraction, care was taken to avoid vortexing and pipetting to prevent shearing of DNA. Where necessary, wide bore tips were used. DNA was extracted with phenol:chloroform; specifically, frozen cell pellets were resuspended in 250 μl D-PBS and 4 volumes of digestion buffer (20 mM Tris HCl pH 8.0, 0.2 M EDTA, 1% SDS, 100 μg/ml DNase-free RNase A) and incubated for 5 min at room temperature. Proteinase K was added to a final concentration of 1 mg/ml and incubated overnight at 56 °C with gentle shaking. Proteinase K addition was repeated until lysates were clear (1–4 h) then an equal volume of phenol:chloroform added. Samples were shaken well and separated by centrifuging at 1700 $g$, 10 min. Phenol:chloroform addition and separation was repeated with the top aqueous layer. An equal volume of chloroform was added to the top aqueous layer, shaken well, and centrifuged as above. DNA was precipitated with 1/10 volume of 7.5 M ammonium acetate and three volumes of ice-cold ethanol or isopropanol and centrifuged at 21,130 $g$ for 1 h at 4 °C. The DNA pellet was washed with 70% ethanol and airdried. DNA was resuspended in 1 × TE overnight at 4 °C. DNA concentration was determined with high sensitivity dsDNA kit for Qubit as per the manufacturer's recommendation (Invitrogen) and 260/230 and 260/280 purity determined with microvolume spectroscopy (Nanodrop or Denovix).

### Ultra-high molecular weight DNA extraction

Ultra-long DNA extractions were performed using the Circulomics Nanobind CBB kit (NB-900–001-01) and UHMW DNA aux kit (provided by the manufacturer on request), following the Circulomics protocol "Nanobind UHMW DNA Extraction

– Cultured Cells Protocol." A pellet containing approximately 6 million cells was used as the sample input for each extraction. For each step that required tip-mixing, the samples were mixed continuously until homogeneous mixtures were achieved. For the overnight elution, a 10-µl pipette tip was left in each tube to ensure that that disc remained submerged in the elution buffer.

### Oxford Nanopore Technology MinION sequencing

Sequencing libraries were prepared using the Genomic DNA by Ligation Sequencing Kit (Oxford Nanopore Technologies, SQK-LSK109) following the manufacturer's instructions with the following changes to enrich for longer read lengths. DNA was incubated at 20 °C for 30 min and 65 °C for 30 min for the end repair step. All AMPure bead cleanup steps used $0.4 \times$ volume of beads and Long Fragment Buffer was used in the final AMPure bead elution wash steps. Sequencing adapter ligations were performed as $0.5 \times$ volume reactions for 30 min at room temperature.

For genome-wide sequencing without barcoding, 4 µg input DNA was used. For genome-wide sequencing with barcoding, 1–1.5-µg input DNA was used with Native barcoding genomic DNA protocol (Oxford Nanopore Technologies) using barcoding kit EXP-NBD104 (Oxford Nanopore Technologies). Barcode ligation reactions were performed as $0.5 \times$ volume reactions and incubated for 30 min at room temperature. Equimolar amounts of barcoded reactions were pooled and 2 µg taken forward for sequencing adapter ligation.

For all sequencing runs, recommended amounts of libraries were loaded onto R9.4.1 MinION flow cells (FLO-MIN106D, Oxford Nanopore Technologies) and sequenced with MinION MkB (Oxford Nanopore Technologies) following the manufacturer's instructions. Where appropriate sequencing runs were paused and flow cells washed and reloaded according to the manufacturer's instructions.

### Oxford Nanopore Technology PromethION sequencing

Sequencing libraries were prepared using the Ultra-Long DNA Sequencing Kit (Oxford Nanopore Technologies, SQK-ULK001) and Nanobind UL Library Prep Kit (Circulomics, NB-900–601-01). DNA input was approximately 40 µg (HeLa-S3) and 15 µg (RPE), both in a volume of 750 µl. For elution, the samples were kept at room temperature overnight and were placed above a magnet to keep the Nanobind disks submerged. The quantity of final library was enough to load the PromethION flow cell three times with nuclease washes in between each load. To maximize sequencing yield, we loaded two R9.4.1 (Oxford Nanopore Technologies, FLO-PRO002) flow cells for each sample and picked the best performing flow cell (in terms of Gb output) to wash and load again. Each flow cell was run on the PromethION 24 for 48 h regardless of whether it was washed and reloaded, with a 6-h pore scan frequency as an optimization for long-read sequencing. Nuclease washes were performed using the Flow Cell Wash Kit (Oxford Nanopore Technologies, EXP-WSH004).

### Nanopore Cas9-targeted sequencing (nCATS)

Cas9 guide RNA sequences and target genomic locations are listed in Additional file 4: Table S3 C. Guides targeting four regions (50–55 kb) were designed using CRISPOR

and CCTop. All guide sequences were cross-checked against previously published HeLa short-read sequencing data [13] for SNPs, and against genome-wide nanopore sequencing data generated in this study to check for structural rearrangements. For Cas9 targeted enrichment libraries, 5–10 μg of input DNA was used. Library preparation followed the Cas9-targeted sequencing protocol and LSK-109 kit from ONT, with an increased ligation reaction time of 3 h. Up to 400 fmol of the prepared library was loaded onto each of two PromethION flow cells.

### BrdU-IP short read sequencing

Genomic DNA, fragmented to 300 bp using a Bioruptor Pico, was prepared for multiplexed pooled anti-BrdU ImmunoPrecipitation Illumina NGS sequencing libraries as [38, 62]. Specifically, starting input for sonication was 6 μg DNA. After sonication, DNA was ethanol precipitated, then underwent End repair and A-tailing using NEBNext Ultra II end repair module (E7546). Illumina compatible primers with barcodes were added using NEBNext Ultra II ligation module (E7595). DNA was purified using AMPure XP beads at 0.9 × then equal quantities of barcoded DNA pooled and 20 ng reserved for input DNA. Three microgam DNA was heat denatured and BrdU-containing DNA immunoprecipitated using 60 μl anti-BrdU antibody (BD, 347,580) in IP buffer (1 × PBS, 0.0625% Triton X-100) overnight at 4 ℃ with rotation. Protein G Dynabeads (60 μl, Thermo Fisher 10003D) were added for 1 h then beads washed three times in ice-cold IP buffer, twice in TE and then eluted in elution buffer (1 × TE, 1% SDS). Immunoprecipitated DNA was purified using AMPure XP beads at 0.9x. IP and input DNA were amplified separately using Illumina compatible indexes and NEBNext Ultra II Q5 Master Mix (M0544) for an equal number of cycles, typically 15–17, depending on recovery, and purified using AMPure XP beads at 0.9 ×.

Libraries were checked for fragment size distribution using Tapestation and libraries quantified using Library Quant as [13]. Libraries were multiplexed where appropriate and at least 35 million reads collected per condition by 80-bp single-end sequencing using NextSeq 500 (Illumina) as [13].

### Mass spectrometry

Mass spectrometry samples were prepared from genomic DNA samples and data collected as described in [38].

### Methylation detection by nanopore sequencing

To examine interference between the signal from CpG methylation and incorporated BrdU, Nanopolish [43] was used to base-call 5 mC using the same sequencing files that were separately used to call BrdU with DNascent2. We visualized methylation and BrdU incorporation on individual reads (e.g., Fig. 1F). Raw nanopore fast5 sequencing files, and subsequent guppy basecalled fastq files, and minimap2 alignment files, generated in the DNAscent pipeline were used to call 5 mC with Nanopolish, using the following commands:

nanopolish -d </path/to/fast5/files/> < corresponding.fastq.gz >

nanopolish call-methylation -r < corresponding.fastq.gz > -b < corresponding_alignment.bam > -g < reference.fasta > > < output.tsv >

### CpG island analysis

CpG island annotations were downloaded from the UCSC Genome Browser [63] and processed bisulfite data from HeLa-S3 cells were downloaded from ENCODE (ENCSR-550RTN) [64]. Bisulfite sequencing BED files that list CpGs, their coverage in the data and the fraction that were found to be methylated (biological duplicates ENCFF696OLO and ENCFF804 NTQ, combined for subsequent analysis) were intersected with the CpG island annotations. CpG islands with sufficient mean coverage ($\geq 3$) in the bisulfite data were ranked by mean proportion methylation, the top third of which were categorized as "high methylation" and the bottom third of which were categorized as "low methylation" (8844 CpG islands, each).

DNAscent detect data (HeLa-S3 cells treated for 20 h with 0, 0.3, 1.5, 5, or 10 μM BrdU) with nanopore read depth and number of BrdU calls at thymidine positions were each converted to bigwig format. The following deepTools [65] commands were used to sum nanopore read coverage and DNAscent BrdU calls in 100-bp windows 2.5 kbp up- and downstream of the CpG island sets described above. Proportion BrdU per 100-bp window were plotted by dividing the summed BrdU count by the read count at thymidine positions.

computeMatrix reference-point –regionsFileName [< lowMeth_CGIs.bed >|< highMeth_CGIs.bed >] –scoreFileName [< nanoDepth.bw >|< DNAscentBrdU.bw >] –outFileName < outMatrix.mat.gz > –referencePoint center –beforeRegionStartLength 2500 –afterRegionStartLength 2500 –binSize 100 –averageTypeBins sum.

### BrdU-IP short read data analysis

Sequencing data were downloaded from Basespace (Illumina), and pre-barcodes were demultiplexed using FASTX barcode splitter:

cat </path/to/fastq/files >| fastx_barcode_splitter.pl –bcfile < text/file/with/barcodes > –bol –prefix < prefix_name >

Pre-barcodes were removed with FASTX barcode trimmer:

fastx_trimmer -f 6 -i < my_file.fastq > -o < my_trimmed_file.fastq.gz > -z.

Sequencing and barcode trimming steps were checked for quality using FASTQC [66]:

fastqc < my_file.fastq > -o </path/to/save/output/>

Reads were mapped to hg38 using BWA-MEM and filtered for uniquely mapping reads and duplicate reads excluded using Samtools. Proportion of BrdU incorporation per sample is calculated as number of uniquely mapped reads (with duplicates excluded) for IP/INPUT, as a proportion of the total pooled sample.

For visualization, coverage at 5′ ends of reads was calculated using bedtools to output a coverage.bed file using the script bwa_map.bash available on our github repository. Blacklist regions were removed using bedtools using the following blacklist; hg38-blacklist.v2.bed   from   https://github.com/Boyle-Lab/Blacklist/tree/master/lists with the addition of two further regions found to have extremely high coverage in our HeLa sequencing data; chr8 127,218,000 127,230,000 and chr15 67,840,000 67,841,000. Coverage was mapped into windows using bedtools. BigWig files were generated for intermediate data visualization using UCSC bedGraphToBigWig using the script gencoverageToBigWig.bash available on our gihub repository.

### Nanopore long-read data analysis

Nanopore sequencing reads were basecalled with guppy and mapped to hg38 with minimap2 [67]. Bam files were filtered where described and indexed with samtools [68] and BrdU incorporation identified with DNAscent2 [39] (first running DNAscent index, then DNAscent detect using the default minimum read length of 1000 bp unless otherwise stated). Detect files were converted to modBAM using the convert_detect_to_modBAM.py script, available in our GitHub repository.

For meta-analysis (plotting of distributions in Fig. 1), BrdU incorporation was analyzed directly from.detect files or after converting to windowed fraction of BrdU incorporation.

### Read visualization

For a read of interest, the relevant line from the modBAM file was converted back to detect format and passed to an R [69] script (read_&_gene_annotation_plotting.R) for visualization. Plots include the probability of BrdU at each thymidine position, the determined level of BrdU incorporation (in windows of 290 thymidines; ~ 1 kb) and where appropriate the inferred replication fork direction, initiation, and termination sites. Finally, the script annotates reads with data from other genomic datasets, including genes from Gencode (GRCh38.p13) [70].

### Identification of fork direction, initiation, and termination sites

Replication fork direction was detected in nanopore sequencing experiments which followed the scheme described in Fig. 2A. Nanopore sequencing and detection of BrdU with DNAscent was carried out as described above. The DNAscent detect data was used to calculate replication fork position and direction and, therefore, replication initiation and termination sites. This process was carried out using a custom R script (ori-ter-fork_calls.R), with major steps outlined below:

(1) Proportion BrdU incorporation was calculated in 290 thymidine windows as described above.

(2) Gradients of BrdU proportion were detected using the Total Variation Regularized Numerical Differentiation algorithm [71]. In short, total variation regularization is used to denoise the first derivative of the windowed BrdU values by fitting a curve which minimizes both regression from the measurements and variance across the fitted curve. Regions of fitted first derivatives greater than 1 indicate rightward forks ($5' \longrightarrow 3'$ on the forward strand of the genome), regions with gradients less than $-1$ indicate leftward forks ($5' \longrightarrow 3'$ on the reverse strand of the genome). These regions, and their orientation, are labelled with open chevrons in the example reads shown.

(3) Adjacent replication fork calls with divergent or convergent orientations were used to define initiation and termination events, respectively.

### Search for sequence motifs at replication initiation sites

The high-resolution DNAscent replication initiation sites (or the subset that intersect with Ok-seq IZs) were analyzed to identify enriched sequence motifs using HOMER [72] as described elsewhere [20]. We did not identify any highly significant sequence motifs and of those identified the most significant were present in < ~ 5% of initiation sites.

**Replication initiation site density (RIGR)**

To compare the observed number of DNAscent replication initiation sites in different genomic regions, we determined the initiation site density, defined as *r*eplication *I*initiations per *G*igabase of mapped *R*eads (abbreviated to RIGR). This controls for any differences in aggregated region size, sequence coverage or ploidy when comparing with a haploid reference genome (Hg38). Briefly, for a particular set of regions (e.g., Ok-seq IZs), the number of intersecting DNAscent initiation sites was determined using bedtools intersect (requiring > 50% of the initiation site to overlap a region of interest). This number of initiation sites was then normalized to the sequence coverage (in Gb) calculated using Samtools bedcov. The significance of observed DNAscent initiation site densities was determined by comparison to 1000 randomized initiation sites. Briefly, each randomization used bedtools shuffle to randomly permute the genomic location of DNAscent initiation sites within a randomly selected subset of the nanopore reads (using Samtools view –subsample).

**Comparison of whole genome DNAscent initiation sites with other datasets**

To compare the locations of DNAscent replication initiation sites with transcribed regions of genome, the high-resolution initiation sites were intersected with annotated genes with support for transcriptional activity. Nucleoplasmic RNA-seq from HeLa-S3 cells [54] was reanalyzed to produce normalized read counts per gene as a measure for RNA polymerase II activity on DNA.

Adapters and low-quality bases were trimmed from raw fastq files using the following cutadapt [73] command:

cutadapt –minimum-length 10 –quality-cutoff 15,10 –trim-n -a AGATCGGAAGAG CACACGTCTGAACTCCAGTCA  -A  AGATCGGAAGAGCGTCGTGTAGGGAAA GAGTGT –output < trimmed_fastq_1.gz > –paired-output < trimmed_fastq_2.gz > < raw_fastq_1.gz > < raw_fastq_2.gz >

Trimmed reads were then pseudo-aligned and gene counts generated using Salmon [74]. Human cDNA, ncRNA, and reference genome (GRCh38.p14) files were downloaded from Ensembl [75], catenated, and the reference genome sequence designated as "decoys." The following Salmon commands were used:

salmon index –transcripts < transcriptome.fa > –index < salmon_index > –decoys < decoys.txt >

salmon quant –libType A –index < salmon_index > –mates1 < trimmed_fastq_1.gz > – mates2 < trimmed_fastq_2.gz > –seqBias –gcBias –posBias –output < salmon_quant >

The output quant.sf file contains normalized read counts (TPM, transcripts per million) for each isoform in the annotated transcriptome. To find the expression levels of genes, the TPM values of all isoforms of a gene were summed. The genomic coordinates of the most highly expressed isoform of a gene were used as the coordinates for the gene. Genes with no read counts were excluded from further analysis. The coordinates of transcribed genes, and their expression level (log2 of TPM), were converted to bigwig format. The following deepTools [65] command was used to plot average transcriptional activity within 100 kbp of DNAscent initiation sites:

computeMatrix reference-point –regionsFileName [< focused_IS.bed >|< dispersed_IS.bed >] –scoreFileName < log2 TPM.bw > –outFileName < outMatrix.mat.gz

> −referencePoint center −beforeRegionStartLength 100,000 −afterRegionStartLength 100,000 −binSize 1000 −averageTypeBins mean.

The output matrices were used to plot heatmaps and were separately processed to find the geometric mean. For significance and to plot 99% confidence intervals, 1000 simulations of DNAscent initiation sites with randomized positions within the mapped reads were analyzed as above.

To assess association with chromatin structure, we compared high-resolution DNAscent initiation sites to DNase-seq rep 1 and 2 (ENCSR959ZXU) and ChIP-seq for the following histone modifications: H2 AFZ ChIP-seq rep 1 and 2 (ENCSR000 AQN), H3 K4 me1 ChIP-seq rep 1 and 2 (ENCSR000 APW), H3 K4 me2 ChIP-seq rep 1 and 2 (ENCSR000 AOE), H3 K4 me3 ChIP-seq rep 1 and 2 (ENCSR340 WQU), H3 K9ac ChIP-seq rep 1 and 2 (ENCSR000 AOH), H3 K9 me3 ChIP-seq rep1 and 2 (ENCSR000 AQO), H3 K27ac ChIP-seq rep 1 and 2 (ENCSR000 AOC), H3 K27 me3 ChIP-seq rep1 and 2 (ENCSR000 APB), H3 K36 me3 ChIP-seq rep 1 and 2 (ENCSR000 AOD), H3 K79 me2 ChIP-seq rep 1 and 2 (ENCSR000 AOG), H4 K20 me1 ChIP-seq rep 1 and 2 (ENCSR000 AOI). The following deepTools [65] command was used to assess chromatin structure 100 kbp up- and downstream of high-resolution DNAscent initiation sites:

computeMatrix reference-point −regionsFileName [< focused_IS.bed >|< dispersed_IS.bed >] −scoreFileName [< DNaseSeq.bw >|< chromatinMarkChIPseq.bw >] −outFileName outMatrix.mat.gz −referencePoint center −beforeRegionStartLength 100,000 −afterRegionStartLength 100,000 −binSize 1000 −averageTypeBins mean.

For significance and to plot 99% confidence intervals, 1000 simulations of DNAscent initiation sites with randomized positions within the mapped reads were analyzed as above.

Relative distance analysis was performed using the BEDTools suite [76] as described previously [77]. Briefly, the relative distance between each DNAscent initiation site and the nearest Ok-seq initiation zone was determined. Statistical significance was determined using 1000 randomizations for the DNAscent initiation site data.

## Fork counts across transcription start sites

To distinguish between inactive genes and highly transcribed genes, expressed genes (identification described above, but excluding overlapping genes) were ranked by their expression level, and divided into two equally sized categories (8123 genes each): "low" and "high" expression.

Replication fork calls, as described above, were separated by directionality, each converted to bigwig format and intersected with a 20-kbp region centered around the transcription start and end sites (TSS and TES, respectively) using the following deepTools command:

computeMatrix reference-point −regionsFileName [< high _TSS.bed >| [< low_TSS.bed >| [< high _TES.bed >| [< low_TES.bed >] −scoreFileName [< left_forks.bw >|< right_forks.bw >] −outFileName outMatrix.mat.gz −referencePoint center −beforeRegionStartLength 10,000 −afterRegionStartLength 10,000 −binSize 10 −averageTypeBins sum.

The columns of the output matrix were summed to get the final forks counts in 10-bp windows with respect to the locations of expressed gene start and end sites. Forks were divided into codirectional or counterdirectional to the direction of gene transcription and plotted using custom R scripts.

## Supplementary Information

---

Additional file 1: Legends for Supplementary Figures and Tables. Fig. S1: Concentration-dependent detection of BrdU in human genomic DNA by nanopore sequencing – further figures, accompanies Figure 1. Fig. S2: Single molecule detection of DNA replication dynamics on ultra-long nanopore sequencing reads – further figures, accompanies Figure 2. Fig. S3: DNAscent reveals stochastic replication initiation sites not identified by population studies – further figures, accompanies Figure 3. Fig. S4: High coverage single molecules at MYC, AFF2 and HBB previously published origins – further figures, accompanies Figure 4. Fig. S5: A fifth of initiation sites are focused by high levels of proximal transcription to define initiation zones – further figures, accompanies Figure 5 [81, 82].

Additional file 2: Table S1: BrdU-IP data.

Additional file 3: Table S2: Fraction nascent reads.

Additional file 4: Table S3: Experimental metrics for single molecule detection of DNA replication dynamics.

Additional file 5: Table S4: DNAscent replication initiation site densityfor various genomic regions.

Additional file 6: Peer review history.

---

### Acknowledgements
The authors thank Stephanie Barker (University of Oxford, for help with reagent preparation), Amanda Williams and Becky Busby (Zoology Sequencing, University of Oxford, for use of NextSeq and Tapestation), Paolo Spingardi and Skirmantas Kriaucionis (University of Oxford, for Mass Spectrometry data collection), Joey Riepsaame (University of Oxford, for help with guide RNA design), and Ildem Akerman (University of Birmingham, for sharing whole-genome datasets).

### Review history
The review history is available as Additional file 6.

### Peer review information

### Authors' contributions
JC, RW, and CN conceived the study and planned the experiments. JC, RW, EdlV, TB, LC, AD, VK, CW, and KG performed the experimental work, JC, RW, EdlV, ST, and CN analyzed the data. JC, RW, and CN wrote the manuscript.

### Funding
This work was supported by the Biotechnology and Biological Sciences Research Council (BBSRC), part of UK Research and Innovation, through the Core Capability Grants BB/CCG1720/1 and BB/CCG2220/1 at the Earlham Institute, its constituents National Capability (BBS/E/T/000PR9816) and Transformative Genomics (BBS/E/ER/23 NB0006), the Earlham Institute Strategic Programme Grant Cellular Genomics BBX011070/1, its constituent work packages BBS/E/ER/230001B (Cellular Genomics WP2 Consequences of somatic genome variation on traits), and grants BB/N016858/1, BB/W006014/1 and BB/Y00549X/1. This work was supported by a Wellcome Trust Investigator Award 110064/Z/15/Z.

### Data availability
Raw (fastq) and processed (bigwig files) BrdU-ip-seq Illumina data are available from NCBI GEO under accession number GSE265956 [78]. BrdU calls on aligned nascent reads (mod.bam format) and inferred replication dynamics (bed files) are available from Zenodo (https://doi.org/10.5281/zenodo.10827586) [79]. Code described here is available from GitHub (https://github.com/DNAReplicationLab/MammalianNanopore) together with a Zenodo repository [80].

## Declarations

### Ethics approval and consent to participate
Not applicable.

### Competing interests
The authors declare that they have no competing interests.

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

## 