## [Additional file 6: Peer review history. · Genome Biology]

Review history

First round of review

Reviewer 1

The paper by Carrington et al reports the use of nanopore sequencing of in vivo BrdU-labeled DNA molecules to provide relatively high-resolution snapshots of replication dynamics in HeLa and RPE human cell lines. This single molecule approach, called DNAscent, was previously established in yeast with previously, well-characterized origin locations, and was able to reveal rarely used sites, which are not detected by population-based approaches. In human cells, methods to reveal origin positions have yielded conflicting results and limited overlap between many different methods. Remarkably, DNAscent finds that most (~80%) initiation sites do not map to known initiation zones but appear to represent stochastic, distributed events. Interestingly, the 20% of sites overlapping with known initiation zones also show similar relationship to TSS sites, suggesting high level transcription might relocalize licensed origins. In contrast, other initiation sites they identify show no relation to transcription levels. No sequence motifs are associated with DNAscent loci. The method is undoubtedly a great addition to the panoply of methods and in principle, brings together several advantages, including base-level resolution, long, single-molecule sequence reads, ease of BrdU labeling and detection.

These are compelling results but somewhat dimmed by possible limited depth of analysis. One aspect that is not completely clear to me and probably requires more discussion is the number of molecules analyzed. Is it only 50 molecules >120kb in length for each cell type "N50>120kb"? How much genome coverage is this? They note that they are sampling the genome so I would expect that few if any regions have been analyzed more than once. If so, there's a lack of data to support the conclusion that dispersed initiation sites are indeed rare. For example, the molecule that shows evidence for three initiation events (Fig. 3B) in a region that OK-seq suggests is replicated in a unidirectional manner in most cells would presumably be very rare. Without multiple factors of genome coverage, this interpretation cannot be tested. It also seems quite surprising that three events appeared to have occurred on the same molecule where each of the three events is seemingly rare based on OK-seq. This result might suggest clustering of origin firing occurs, contrary to conclusion of the optical mapping study.

If correct that only 50 molecules were analyzed, the number of origins identified seems like a high density. Is it more than other studies? If population-based approaches only capture 20% of initiations, this study should identify 5x as many per kbp.

The fork direction analysis and calling of initiation and termination regions appears ambiguous in many regions. Dips in BrdU incorporation are sometimes called as origins, but often not. Higher levels of genome coverage would presumably help resolve these cases for sites used with some frequency. However, as most sites are used infrequently, they can only be validated with much higher coverage. Using the low coverage data, therefore, will generally not reflect the replication kinetics of most genomes as yielded by OK-seq for example. This may limit the application of this approach for deep analysis of specific site(s). Perhaps ironically, the paradigmatic finding that 80% of sites are stochastic and dispersed, suggests that mapping origins is futile.

Figure S4: are the x-axes labels correct in B and C?

Reviewer 2

To investigate the nature of replication initiation sites in human cells, Carrington et al developed a method to analyse BrdU-incorporated replication tracks using nanopore sequencing technology, enabling mapping of replication tracks at a single molecule level. In terms of this technological development, the authors carefully examined experimental conditions as well as analytical steps (such as the concentration and labelling time of BrdU, discrimination of BrdU vs. CpG methylation, etc.) and successfully detected replication tracks with high resolution. While the throughput level of this method is not sufficient to profile replication of the whole genome, it successfully detected many replication initiation sites that were not detected in population-based methods. The authors term them 'dispersed' initiation sites, which occur at variable locations in the genome. This is quite surprising but consistent with replication of inter-origin zones (IZs), some of which span hundreds of kilobases to several megabases. Although the nature of 'dispersed' initiation sites has not been fully elucidated yet, this study considerably advances our understanding of the stochastic nature of DNA replication initiation and is therefore suitable for publication. I've listed comments on specific points below:

- To identify replication tracks, BrdU was consecutively added to the cell culture with increasing concentrations. However, the authors did not clarify why such treatment is required. I suggest the authors clarify this point. For example, I wonder if a one-off BrdU labelling with or without subsequent wash could also generate a gradient of BrdU incorporation during the course of replication.

- Authors performed BrdU-IP-seq and compared the amount of pulled-down nascent DNA with the level of BrdU quantified by mass spectrometry. However, I do not think that this comparison is meaningful in this manuscript and this part could be deleted.

- On page 12, in the second paragraph, authors describe the number of replication tracks detected by DNAscent. However, they did not show how broadly these tracks are distributed across the genome. I suggest the authors add a figure to show an overview genome-wide mapping pattern (or mapping frequencies) of replication tracks.

- On page 15, line 9, 'In contrast to Ok-seq, Pu-seq and ORM IZ~ or ini-seq[23,24].' I think this sentence is not accurate. As I understand it, OK-seq, Pu-seq and ORM IZ do not colocalize well with ini-seq (v1) or sns-seq-detected initiation sites.

- On page 16, lines 9-10, 'co-directional replication forks are overrepresented from 3 kb upstream ~at least 10 kb into the gene body.' Is this bias of fork directionality only observed around the transcription start site (TSS) or is it a uniform feature in gene bodies? To distinguish these possibilities, I suggest the authors analyse fork directionality around transcribed regions categorized by gene length.

- Given the lack of any genomic features (sequence, chromatin status) at 'dispersed' initiation sites (Fig. S4D), I am wondering if, in a small fraction of cells, 'dispersed' initiation events also occur at predetermined genomic regions or if these are a readout for completely random initiation events across the genome. Can the authors comment on this point?

- In the discussion, some parts repeat the content of the results section, which may not be necessary.

Reviewer 3

The location of replication origins in vertebrate cells has been actively studied for decades, but genomics methods using population-based assays have often generated conflicting results. In this study the Nieduszynski group employed Nanopore sequencing with BrdU detection (DNAscent) to map replication events with single-molecule precision. Building on work in budding yeast, they employ a strategy to gradually increase BrdU labeling over a defined period. This allows the directionality of DNA replication to be inferred from a BrdU gradient in the mapped data. With this information, Carrington et al. are able to define sites of replication initiation at many sites across the genome. In addition, they confirm that their data typically coincides with OK-seq at well-defined origins. However, defined origins represent a minority of replication initiation sites within the genome, the authors then show that Nanopore sequencing can identify many previously unmapped replication origins.

Overall, this work represents an important addition to our understanding of how the human genome is replicated, the establishment of DNAsent/Nanopore method in human cells and the work done to validate the BrdU signals in Nanopore is of high importance. The potential impact of this work is lessened by technical limitations: most significantly, the throughput of the Nanopore is very low so that origin mapping is anecdotal. Thus, statistics related to the prevalence of origins mapped to individual locations is unavailable. In addition, the nature of the BrdU pulse, performed in asynchronous cells, makes it very difficult to precisely map initiation sites.

Major points:

The manuscript could be significantly improved if the authors could generate data that maps the same origin(s) with multiple independent reads - while this is difficult to accomplish, perhaps the authors could trial adaptive sampling to focus on a specific region.

In OK-seq data large regions of the genome appear unidirectionally replicated, however given the size of these regions, the replication rate and the length of S-phase, these must contain many dispersed origins. With that in mind, the findings presented here are not surprising. The data shown in figure 3B shows clustered bi-directional initiation events, but the corresponding OK-seq data implies that replication occurs in a dominant orientation. The authors could investigate this apparent discrepancy in closer detail; in particular, I believe that the OK-seq data is presented as a ratio of signal mapped to the two DNA strands, which can give the erroneous impression that DNA replication is unidirectional. The authors could look at raw OK-seq data mapped to each of the strands and then specifically look for patterns of bidirectional replication in these regions.

The figures show initiation events of different sizes, which is related to the initial detection of the BrdU pulse, however the edge of the initiating event does not align with the start of BrdU signal. Perhaps the authors could better annotate how certain initiation sites are chosen.

Authors' response to reviewers

Response to reviewers

(reviewer comments in black italics; responses in blue; quotes from manuscript in blue italics)

We thank all three reviewers for their positive reviews of our manuscript and their valuable suggestions to improve the work. We address each of their points in turn below:

Reviewer #1:

These are compelling results but somewhat dimmed by possible limited depth of analysis. One aspect that is not completely clear to me and probably requires more discussion is the number of molecules analyzed. Is it only 50 molecules >120kb in length for each cell type "N50 >120 kb"? How much genome coverage is this?

We apologise for the confusion - our text describing N50 was unclear. The N50 metric describes a value where half of the data is contained within reads with lengths greater than this.

Therefore, for the value quoted by the referee, half of the data we collected is contained within reads >120 kb in length.

To clarify this we have updated the manuscript text to include the following (page 11):

“(half of the data contained in reads longer than (N50) 120 kb; Supplementary Table S3A)”.

In addition, we have updated supplementary table S3 to include read metrics from these runs. For example, for the dataset collected from HeLa-S3 cells, we sequenced over 1.3 million reads of which 22,387 contained BrdU. We identified replication forks on 13,064 reads and replication initiation events on 2,256 reads. In total we identified 2,577 and 912 replication initiation events in the HeLa-S3 and hTERT-RPE1 cell datasets, respectively.

They note that they are sampling the genome so I would expect that few if any regions have been analyzed more than once. If so, there's a lack of data to support the conclusion that dispersed initiation sites are indeed rare.

We thank the reviewer for their comments and agree that our manuscript would benefit from high coverage data for some areas of the genome. To achieve this, we have performed additional experiments by combining our DNAscent methodology with enrichment for four specific genomic locations using nanopore Cas9-targeted sequencing (nCATS). These additional data are included in new figure 4 (and associated new supplementary figure S4) and associated results sections. The selected loci include two that meet our definition for focused initiation sites (proximal to the MYC and TOP1 genes respectively) and two with anticipated dispersed initiation sites (proximal to the HBB and AFF2 genes). The nCATS enrichment method generated nascent strand (defined as reads containing BrdU) coverage of 125 and 242 for TOP1 and MYC (respectively) and allowed us to identify 14 and 19 initiation events in proximity to TOP1 and MYC respectively (RIGR = 103.2 and 84.8). We generated nascent strand coverage of 29 and 99 for AFF2 and HBB respectively. In contrast to the density of initiation events proximal to TOP1 and MYC, we identified only 1 and 3 initiation events in proximity to AFF2 and HBB respectively (RIGR = 16.4 and 19.6). This is consistent with individual sites rarely being used for dispersed initiation events by comparison to the more frequent use of focused initiation sites. We have added discussion of the relative expected density of initiation sites across focussed or dispersed regions to the discussion.

For example, the molecule that shows evidence for three initiation events (Fig. 3B) in a region that OK-seq suggests is replicated in a unidirectional manner in most cells would presumably be very rare. Without multiple factors of genome coverage, this interpretation cannot be tested. It also seems quite surprising that three events appeared to have occurred on the same molecule where each of the three events is seemingly rare based on OK-seq. This result might suggest clustering of origin firing occurs, contrary to conclusion of the optical mapping study.

While we anticipate that specific initiation sites identified in Fig. 3B would indeed be rarely used across a population of cell, it is still necessary for the cell to initiate replication somewhere in the vicinity of the region where this read maps. With average inter-origin distances measured at

~100 kb and the read spanning 234 kb, it is likely that there is an initiation event somewhere across this region in most cells. To reconcile this with the lack of an initiation zone in Ok-seq data we propose that the initiation site used varies between cells (is dispersed) and is therefore 'averaged out' and not visible in cell population data.

We have added the following to the discussion (page 25):

“Whilst large regions of the genome appear unidirectionally replicated in cell population level Ok-seq data, we identify initiation events within these regions. Given that these regions are often much larger than the average IOD of ~100 kb, this is perhaps unsurprising. We suggest that at a single cell/molecule level initiation occurs within these regions at the expected average of once per ~100 kb but that these sites are dispersed such that when analysing cell population data, the various initiation events are ‘averaged out’ with little to no preference for initiation at particular sites (i.e. no IZ or focussed initiation sites).”

If correct that only 50 molecules were analyzed, the number of origins identified seems like a high density. Is it more than other studies? If population-based approaches only capture 20% of initiations, this study should identify 5x as many per kbp.

Please see above for an expanded definition of N50 and the number of reads analysed. We report 2,577 initiation events across 22,387 nascent (BrdU containing) reads in the HeLa-S3 dataset. Comparison of density with existing population studies is complicated. Existing population studies vary in the number of identified origins by several orders of magnitude. Our main conclusion is the observation that 80% of the initiation events we identify do not colocalise with sites identified by cell population-based methods. Whilst initiation events that occur together on one single molecule clearly co-occurred within the same cell, neighbouring initiation sites on different molecules are unlikely to come from the same cell (given that the sample originated from ~10 million cells from which 22,387 nascent molecules were identified – i.e. ~1 molecule per 450 cells).

The fork direction analysis and calling of initiation and termination regions appears ambiguous in many regions. Dips in BrdU incorporation are sometimes called as origins, but often not.

We agree with the reviewer that there are some dips in BrdU signal that are not identified as initiation sites. We intentionally set a high stringency for replication fork, initiation and termination event calling to favour specificity (low false positive rate) at the expense of some sensitivity (true positive rate). We made this choice to have high confidence in the initiation sites that are identified. Our use of an unbiased approach for gradient and initiation site (and termination site) identification allows consistent identification across the genome. Therefore, in our study, due to the unbiased, stringent algorithm for initiation event detection, we can be highly confident in the initiation events detected.

We have added the following sentence to the results (page 11):

“We intentionally set a high stringency for initiation site calling to favour specificity (low false positive rate) over some sensitivity (true positive rate), to have high confidence in the initiation sites that are identified.”

Higher levels of genome coverage would presumably help resolve these cases for sites used with some frequency. However, as most sites are used infrequently, they can only be validated with much higher coverage. Using the low coverage data, therefore, will generally not reflect the replication kinetics of most genomes as yielded by OK-seq for example. This may limit the application of this approach for deep analysis of specific site(s). Perhaps ironically, the paradigmatic finding that 80% of sites are stochastic and dispersed, suggests that mapping origins is futile.

The reviewer raises interesting and important points:

1. With regard to coverage, we agree with the reviewer, and this contributed towards our motivation to generate higher coverage data in the new target enrichment experiments. These new data clearly demonstrate that many more initiation events are seen at the selected focused initiation sites than at the dispersed initiation sites. Despite the challenge we were able to obtain sufficient coverage to identify replication initiation events over the two dispersed initiation sites present in our target enrichment data.

2. Regarding the utility of the methodology, it is important to recognise that optical detection of nucleotide analogues with methods such as DNA combing and fibre analysis have elucidated many important insights into chromosome biology. The methodology that we introduce offers significant advantages over these optical methods, including higher throughput, higher resolution, less subjective data interpretation and (perhaps most importantly) the underlying sequence information. Therefore, we are confident that there will be many future applications of this technology.

Figure S4: are the x-axes labels correct in B and C?

We thank the reviewer for pointing out this error which we have corrected.

Reviewer #2:

To identify replication tracks, BrdU was consecutively added to the cell culture with increasing concentrations. However, the authors did not clarify why such treatment is required. I suggest the authors clarify this point. For example, I wonder if a one-off BrdU labelling with or without subsequent wash could also generate a gradient of BrdU incorporation during the course of replication.

We thank the reviewer for raising this interesting point. We undertook experiments with one-off additions (shorter pulses but treatment as described in Fig. 1) and observed rapid transitions from no BrdU to high BrdU incorporation (from background to >60% substitution) with 1-2 kb. We also found similarly rapid transitions from high to low BrdU incorporation when combining a single BrdU concentration treatment with a subsequent wash. The resulting data contained a series of sharp transitions from low to high BrdU from which it was challenging to identify the direction of replication forks and the sites of replication initiation. By comparison the sequential addition regime that we employed results in a slower increase in BrdU incorporation giving the ability to call fork direction and giving confidence to identified replication initiation events. Finally, the sequential additions (over 1 hour) allow us to resolve a wider temporal window than would be possible with a single BrdU addition.

We have added the following sentences into the results section (page 11):

“This regime of BrdU additions to cells generated regions of increasing BrdU incorporation across sequencing reads from which we could calculate gradients and fork directions. In contrast single pulse treatments of BrdU resulted in rapid transitions from low to high BrdU incorporation (within 1-2 kb) and consequently lower temporal resolution and reduced ability to identify fork direction (data not shown).”

Authors performed BrdU-IP-seq and compared the amount of pulled-down nascent DNA with the level of BrdU quantified by mass spectrometry. However, I do not think that this comparison is meaningful in this manuscript and this part could be deleted.

We feel that the comparison of our single molecule BrdU detection data to a well-established (cell population BrdU-IP-seq) method is meaningful and offers important validation.

On page 12, in the second paragraph, authors describe the number of replication tracks detected by DNAscent. However, they did not show how broadly these tracks are distributed across the genome. I suggest the authors add a figure to show an overview genome-wide mapping pattern (or mapping frequencies) of replication tracks.

We thank the reviewer for this valuable suggestion. We have added two panels to one of our supplementary figures (Fig. S2A & B) to visualise the sequence coverage of BrdU containing reads (nascent molecules) and the location of identified replication initiation events from the genome-wide datasets (HeLa-S3 and hTert-RPE1 cells). These figures confirm that the nascent molecules and replication initiation sites are found across the whole genome.

On page 15, line 9, 'In contrast to Ok-seq, Pu-seq and ORM IZ~ or ini-seq[23,24]. I think this sentence is not accurate. As I understand it, OK-seq, Pu-seq and ORM IZ do not colocalize well with ini-seq (v1) or sns-seq-detected initiation sites.

We agree that this sentence was poorly worded and open to misinterpretation. We have rephrased it to (page 15):

“When considering all high-resolution (<5 kb) DNAscent initiation sites, we observed significant enrichment with Ok-seq, Pu-seq and ORM IZs (Fig. 2D), but not with the published sites identified by chromatin immunoprecipitation (ChIP; ORC [26] or Mcm7 [27]), SNS-seq [46], or Ini-seq [23, 24] (Supplementary Fig. S3A).”

On page 16, lines 9-10, 'co-directional replication forks are overrepresented from 3 kb upstream ~at least 10 kb into the gene body.' Is this bias of fork directionality only observed around the transcription start site (TSS) or is it a uniform feature in gene bodies? To distinguish these possibilities, I suggest the authors analyse fork directionality around transcribed regions categorized by gene length.

We thank the reviewer for raising this interesting question and suggesting a valuable analysis. We have sub-categorised genes into bins based upon expression level (high and low) and gene length (four groups); these data are presented in Supplemental Fig. S5D. As anticipated, we see no bias in fork direction across any of the gene size sets for the poorly expressed genes. For the highly expressed gene sets the bias for co-directional replication and transcription is strongest in the longest genes where it extends at least 10 kb beyond the TSS. We refer to this additional analysis in our revised manuscript.

Given the lack of any genomic features (sequence, chromatin status) at 'dispersed' initiation sites (Fig. S4D), I am wondering if, in a small fraction of cells, 'dispersed' initiation events also occur at predetermined genomic regions or if these are a readout for completely random initiation events across the genome. Can the authors comment on this point?

We thank the reviewer for this interesting comment. Dispersed initiation events may be at predetermined genomic regions, but (as the reviewer suggestions) in such a small fraction of cells that they cannot be detected by population replication dynamics (e.g. Ok-seq) approaches. Furthermore, it is possible that in the single cell that a particular dispersed initiation event took place the site was 'predetermined', for example, by a permissive chromatin environment.

We have added the following to the discussion:

“We suggest this is due to the probability of these sites firing within each cell, and therefore across a population, being comparatively low. We do not identify any common chromatin feature (from population data) between dispersed initiation sites, but it remains possible that such a signature is present in the single cell that a particular dispersed initiation event took place. Nevertheless, we cannot exclude the possibility that these dispersed sites are largely random.”

In the discussion, some parts repeat the content of the results section, which may not be necessary.

We have edited the discussion to remove repetition with the results section.

Reviewer #3:

The potential impact of this work is lessened by technical limitations: most significantly, the throughput of the Nanopore is very low so that origin mapping is anecdotal. Thus, statistics related to the prevalence of origins mapped to individual locations is unavailable. In addition, the nature of the BrdU pulse, performed in asynchronous cells, makes it very difficult to precisely map initiation sites.

We thank the reviewer for raising these interesting points.

Throughput: In the original manuscript, our application of the DNAscent technology is analogous to other single molecule/cell approaches which sample across a genome rather than comprehensively covering the whole genome at depth (as cell population technologies may). Of course, our use of nanopore sequencing offers dramatic improvements over current single molecule methods (such as combing/fibre analysis) as we discuss in our response to the final point from reviewer 1. For example, while sampling across the genome doesn't allow statistical analysis at individual locations the increased throughput of DNAscent relative to combing/fibre methods combined with the underlying sequence data does permit powerful genome-wide meta-analyses (e.g. Fig. 5). This allowed our discovery that most of the human genome is replicated from sites missed by Ok-seq analysis.

Individual locations: we agree that increased coverage over individual locations would increase the impact of our work and this in part motivated the target enrichment experiments included in this revised manuscript. These data have allowed statistical analysis of the distribution of replication initiation sites at the selected individual locations. This demonstrates that the DNAscent technology can be applied in two powerful and complementary modes: genome-wide sampling and high-coverage at specific targeted regions of interest.

Precision of initiation site mapping: we recognise that there is a fundamental limit to the resolution of initiation site mapping from our approaches. This is the case for all approaches that map replication initiation sites based upon experimental determination of replication dynamics. However, we note that our mapping of sites of replication initiation within ~1 kb in proximity to TOP1 and MYC (Fig. 4 and S4a) offer an order of magnitude improvement over current Ok-seq data (IZs at TOP1 and MYC from Ok-seq data span 55 kb and 66 kb respectively). Our genome-wide analyses of replication initiation sites was restricted to those sites that we

could identified <5 kb resolution – again offering a substantial improvement in resolution over Ok-seq and ORM analyses.

The manuscript could be significantly improved if the authors could generate data that maps the same origin(s) with multiple independent reads - while this is difficult to accomplish, perhaps the authors could trial adaptive sampling to focus on a specific region.

We thank the reviewer for this valuable suggestion. Please see the response above to reviewer 1 and the addition of targeted enrichment data to our manuscript.

In OK-seq data large regions of the genome appear unidirectionally replicated, however given the size of these regions, the replication rate and the length of S-phase, these must contain many dispersed origins. With that in mind, the findings presented here are not surprising. The data shown in figure 3B shows clustered bi-directional initiation events, but the corresponding OK-seq data implies that replication occurs in a dominant orientation. The authors could investigate this apparent discrepancy in closer detail; in particular, I believe that the OK-seq data is presented as a ratio of signal mapped to the two DNA strands, which can give the erroneous impression that DNA replication is unidirectional. The authors could look at raw OK-seq data mapped to each of the strands and then specifically look for patterns of bidirectional replication in these regions.

We thank the reviewer for the insightful comment and useful suggestion. We have examined raw Ok-seq data mapped to each strand. For example, considering the region shown in Fig. 3B, in the raw Ok-seq data there is substantial variability in how many reads map to each 1 kb bin (combining both strands), the number of reads mapped to each strand and in the resulting replication fork direction. It is not possible to distinguish technical (e.g. from the DNA amplification required before high-throughput sequencing) from biological causes (e.g. dispersed initiation sites) for this variability at 1 kb length scales. Consequently Ok-seq data analyses look at the change in the ratio of signal mapped to the two DNA strands (i.e. the first differential) rather than the absolute values.

We have added the following to the discussion (page 25):

“Whilst large regions of the genome appear unidirectionally replicated in cell population level Ok-seq data, we identify initiation events within these regions. Given that these regions are often much larger than the average IOD of ~100 kb, this is perhaps unsurprising. We suggest that at a single cell/molecule level initiation occurs within these regions at the expected average of once per ~100 kb but that these sites are dispersed such that when analysing cell population data, the various initiation events are ‘averaged out’ with little to no preference for initiation at particular sites (i.e. no IZ or focussed initiation sites).”

The figures show initiation events of different sizes, which is related to the initial detection of the BrdU pulse, however the edge of the initiating event does not align with the start of BrdU signal. Perhaps the authors could better annotate how certain initiation sites are chosen.

Our approach involved first determining the location of gradients in BrdU incorporation to infer replication fork direction. Subsequently, divergent bi-direction replication fork annotations were used to infer initiation sites (as performed in DNA combing/fibre experiments). Therefore, our approach is founded on determining gradients in BrdU incorporation. In essence, gradients can only be calculated by comparison between adjacent data positions, hence the partial extension of replication fork calls into regions of low BrdU incorporation.

We have added the following sentence to the results (page 11):

“We intentionally set a high stringency for initiation site calling to favour specificity (low false positive rate) over some sensitivity (true positive rate), to have high confidence in the initiation sites that are identified.

Second round of review

Reviewer 1

The authors’ responses have addressed my concerns and added clarity. Nice work

Reviewer 2

The authors adequately responded to all the comments and fully addressed the issue I was concerned about. The revised manuscript can be published.

Reviewer 3

The authors have generally addressed my previous critiques and the manuscript is improved. In particular the addition of the targeted enrichment data has allowed the authors to provide a more detailed analysis of select sites of replication initiation. Some issues do remain with resolution of the origin calls and the fact that only a subset of the enriched reads are truly informative for origin mapping. Nevertheless, some important aspects of DNA replication are revealed and this work is informative in the long-standing quest to understand DNA replication in mammalian cells.

Minor comments:

Could the authors show a larger window in figure 4B (and equivalents in the supplement) as the box seems to arbitrarily cutoff a lot of data? Also, for clarity can the heatmaps be organized by origin position?

Dark vs light green for termination or initiation is difficult to distinguish.

Add annotations/key to figure 5 to explain line color representation.